# Antiparasitic Activity of Tea Tree Oil (TTO) and Its Components against Medically Important Ectoparasites: A Systematic Review

**DOI:** 10.3390/pharmaceutics14081587

**Published:** 2022-07-29

**Authors:** Solomon Abrha Bezabh, Wubshet Tesfaye, Julia K. Christenson, Christine F. Carson, Jackson Thomas

**Affiliations:** 1Faculty of Health, University of Canberra, Bruce, Canberra, ACT 2617, Australia; solomon.bezabh@canberra.edu.au (S.A.B.); wubshet.tesfaye@canberra.edu.au (W.T.); julia.christenson@canberra.edu.au (J.K.C.); 2Department of Pharmaceutics, School of Pharmacy, College of Health Sciences, Mekelle University, Mekelle 7000, Ethiopia; 3School of Pharmacy, Faculty of Medicine and Health, University of Sydney, Camperdown, NSW 2050, Australia; 4School of Medicine, The University of Western Australia, 35 Stirling Highway, Crawley, WA 6009, Australia; christine.carson@uwa.edu.au

**Keywords:** antiparasitic, *Demodex* mites, ectoparasites, fleas, house dust mites, lice, scabies mites, tea tree oil (TTO), TTO components

## Abstract

Ectoparasites are pathogens that can infect the skin and cause immense pain, discomfort, and disease. They are typically managed with insecticides. However, the fast-emerging antimicrobial resistance and the slow rate of development of new bio-actives combined with environmental and health concerns over the continued use of neurotoxic insecticides warrant newer and alternative methods of control. Tea tree oil (TTO), as an alternative agent, has shown remarkable promise against ectoparasites in recent studies. To our knowledge, this is the first systematic review to assess preclinical and clinical studies exploring the antiparasitic activity of TTO and its components against clinically significant ectoparasites, such as *Demodex* mites, scabies mites, house dust mites, lice, fleas, chiggers, and bed bugs. We systematically searched databases, including PubMed, MEDLINE (EBSCOhost), Embase (Scopus), CENTRAL, Cochrane Library, CINAHL, ScienceDirect, Web of Science, SciELO, and LILACS in any language from inception to 4 April 2022. Studies exploring the therapeutic activity of TTO and its components against the ectoparasites were eligible. We used the ToxRTool (Toxicological data reliability assessment) tool, the Joanna Briggs Institute (JBI) critical appraisal tools, and the Jadad scale to assess the methodological qualities of preclinical (in vitro and in vivo) studies, non-randomised controlled trials (including cohort, case series, and case studies), and randomised controlled trials, respectively. Of 497 identified records, 71 studies were included in this systematic review, and most (66%) had high methodological quality. The findings of this review revealed the promising efficacy of TTO and its components against ectoparasites of medical importance. Most importantly, the compelling in vitro activity of TTO against ectoparasites noted in this review seems to have translated well into the clinical environment. The promising outcomes observed in clinical studies provide enough evidence to justify the use of TTO in the pharmacotherapy of ectoparasitic infections.

## 1. Introduction

Neglected tropical diseases (NTDs) are a group of communicable diseases that affect nearly two billion people worldwide and kill over 500,000 people annually [1,2]. They are endemic to impoverished communities living in low- and middle-income countries (LMICs), and are increasingly being recognised as the emerging causes of cardiovascular diseases (CVDs) in these countries [1,3]. CVDs are the leading cause of death worldwide, and >80% of these deaths occur in LMICs, with rheumatic heart disease (RHD) remaining a substantial preventable cause of cardiovascular disability and death [4,5]. About 95% of RHD cases occur in LMICs [4]. Scabies, myasis, tungiasis, and other ectoparasites have also been added recently to the global NTD portfolio [6]. 

Ectoparasites are pathogens that usually infect the skin of humans or other host organisms [7]. While temporarily blood-sucking arthropods (e.g., mosquitoes) are considered ectoparasites, the term is mainly used to refer to parasites such as mites, lice, fleas, and bedbugs that live on or in the skin [7,8]. Ectoparasites can cause serious diseases either directly by sucking blood or indirectly as vectors of infectious diseases, collectively posing a serious threat to human health and a significant burden to the global economy [7,9]. Among ectoparasitic diseases, scabies, demodicosis, headlice, and tungiasis are known as ectoparasitic diseases of medical importance as they cause substantial human morbidity [7,8]. Ectoparasitic diseases can be sporadic, endemic, or epidemic, depending on the type and place of living [10]. For example, in Australia, although the prevalence of scabies in the general population is low, the condition is hyperendemic in rural remote Aboriginal communities [11]. Similarly, about 80% of vulnerable children from Kenya and almost all indigenous peoples in the Amazon rainforest are impacted by tungiasis and head lice, respectively [10,12].

Over the years, several insecticides and pesticides have been successfully used to treat ectoparasitic infestations; however, as with other antimicrobial agents, overuse of these agents has led to the development of resistance, which is a worrisome public health concern [13,14]. As a result, screening plant products, with a key focus on secondary plant metabolites such as essential oils (EOs), has become important in the search for alternative therapeutic solutions [15,16,17,18,19]. EOs have traditionally been used for centuries for the treatment of ectoparasitic infestations—this is because of their antiparasitic, antibacterial, and/or anti-inflammatory properties [14,20]. However, most EOs have weak to moderate antimicrobial activities and are overshadowed by more active synthetic agents in practice [16]. In fact, only a few of them produce broad activity against a wide range of microbes. Tea tree oil (TTO), the EO obtained from *Melaleuca alternifolia*, is one such EO with potent and broad antimicrobial properties [16,21,22].

TTO contains approximately 100 compounds. Among the components of TTO, terpinen-4-ol (T4O) γ-terpinene, α-terpinene, 1,8-cineole, and terpinolene are the main bioactive, and most abundant, components. T4O and α-terpineol have been identified as the components most responsible for TTO’s antimicrobial activity. These components have been standardised for TTO quality control by the industry, as per the International Organization for Standardization standard (ISO 4730) [21,22]. TTO possesses a unique combination of potent acaricidal, insecticidal, antibacterial, wound healing, antioxidant, and anti-inflammatory effects [22]. As a result, it has long been explored as a topical treatment for a variety of ectoparasite infestations, including head lice, scabies, and demodicosis, with good safety and efficacy data [22,23]. It is known for its potent activity as a bactericide (at 0.002–2%), including against methicillin-resistant *S. aureus* (MRSA), and as an anti-inflammatory agent (≤0.125%). Bacterial secondary infection and inflammation are both often associated with ectoparasitic infections [21,22]. The leaves of *Melaleuca alternifolia* have been used as bush medicine for different skin diseases by Australian Aboriginals, and the steam distilled oil has been used widely by Australian communities for more than 90 years [23]. TTO is an active ingredient in products registered in the UK’s Medicines and Healthcare products Regulatory Agency and listed on the Australian Register of Therapeutic Goods.

The mechanism by which TTO produces its antiparasitic effect has not been fully elucidated. However, its miticidal effect is partly attributed to the anticholinesterase activity of T4O, 1,8-cineole, γ-terpinene, α-terpinene, and ρ-cymene, which can cause lethal muscular contraction and spastic paralysis of the parasite (Figure 1) [24,25,26].

TTO’s anticholinesterase inhibition is shown to be more potent than that of the individual components [26], suggesting a synergistic effect of the components responsible for its antiparasitic activity [28,29,30]. The combined action of multiple active ingredients may reduce the potential for development of resistance to TTO, as multiple simultaneous mutations would be required to overcome all the actions of the individual components [22]. In lice, TTO is shown to cause bulging of respiratory spiracles that might lead to suffocation (Figure 2) [31].

Given ectoparasite infestations progress to inflammatory skin reactions and secondary bacterial complications [9,13,33], TTO could be a good fit in managing associated co-morbidities and secondary complications, attributed to its anti-inflammatory, antimicrobial, and wound-healing properties.

While several systematic reviews [34,35] and narrative reviews [21,36,37,38,39] have explored the antibacterial, anti-inflammatory, antifungal, and antiviral activities of TTO, few have comprehensively investigated its antiparasitic activity. One narrative review [29] summarised the studies evaluating TTO against *Demodex* mites and five [28,30,40,41,42] systematically reviewed clinical studies assessing TTO and other anti-*Demodex* agents. To our knowledge, this is the first systematic review of preclinical (in vitro and in vivo) and clinical studies exploring TTO and its components against medically important ectoparasites, including mites (*Demodex*, scabies, and house dust), lice, fleas, chiggers, and bed bugs. These ectoparasites cause extensive morbidity to humans by either directly feeding on the host or causing allergic reactions and other serious diseases [7,9]. A review of this nature can help establish the evidence base for the efficacy and safety of TTO and its components against these ectoparasites, and inform clinical practice and direct future studies in this space.

## 2. Materials and Methods

### 2.1. Study Design

Initial searches revealed that published studies varied considerably in terms of study interventions, duration of treatment, participants, study design, study outcome measures, and follow-up durations, making a meta-analysis impossible. Hence, narrative-style data synthesis was employed to systematically organise, present, and appraise preclinical and clinical data.

### 2.2. Search Strategies and Selection Criteria

This systematic review was registered with the International Prospective Register of Systematic Reviews (PROSPERO; CRD42020212037) and is reported according to the Preferred Reporting Items for Systematic Reviews and Metanalyses (PRISMA) statement (Appendix A, pp. 3–4) [43]. Two researchers (S.A.B. and W.T.) independently searched for in vitro, in vivo, and clinical studies exploring the use of TTO against the selected medically important ectoparasites using combinations of the terms “tea tree oil”, “*Melaleuca alternifolia* oil”, parasites, “ectoparasitic infestations”, mites, “mite infestations”, scabies, blepharitis, Pyroglyphidae, Trombiculidae, *Pediculus*, “lice infestations”, Phthirapteran, flea, “flea infestations”, Siphonaptera, *Tunga*, tungiasis, and “bed bugs”. The databases searched were PubMed, MEDLINE (EBSCOhost), Embase (Scopus), CENTRAL (Cochrane Central Register of Controlled Trials), Cochrane Library, CINAHL (Cumulative Index to Nursing & Allied Health Literature), ScienceDirect, Web of Science, SciELO (The Scientific Electronic Library Online), and LILACS (Latin America and Caribbean Health Sciences Literature). Searches were performed without language restrictions from database inception to 12 November 2020. The search was then updated on 4 April 2022, using the same search terms and 12 new records targeting *Demodex* mites (in vitro (*n* = 2), randomised controlled trials (RCTs, *n* = 4), quasi-experimental (*n* = 2), cohort, case series, and cases studies (*n* = 1 each)) were identified and included in the review. The full search strategy is summarised in Appendix A, pp. 1–2). Grey literature was searched in Australian Tea Tree Industry Association (ATTIA) database. Additional searches were performed in Google and Google Scholar, and reference lists of included papers were manually screened to target articles potentially missed during the main search.

To perform the screening, the records obtained from the search results were exported to Covidence (Veritas Health Innovation, Melbourne, Australia) [44]. After duplicates were removed, two researchers (S.A.B. and W.T.) independently screened the titles and abstracts of the records for relevance and reviewed the full-text articles for eligibility. Any disagreements between the two researchers were resolved via discussion. Articles published in languages other than English were translated by Google Translate.

All in vitro, in vivo, and clinical studies with either interventional or observational designs reporting the antiparasitic effects of TTO and/or its components or combination of TTO and/or its components with other treatments against ectoparasites of medical importance, such as mites (i.e., *Demodex* mites, scabies mites, house dust mites, chiggers mites), lice, fleas, and bed bugs were considered in this review. Reviews were excluded along with TTO studies on antibacterial, anti-inflammatory, antioxidant, antifungal, antiviral, antiprotozoal effects, endo-parasites, and ectoparasites of veterinary importance, including cattle mites, donkey lice, sheep lice, cattle tick, pig mites, and dog *Demodex* mites. Although excluded from the review, the records reporting TTO and its components against veterinary important ectoparasites were summarised to give a comprehensive antiparasitic profile (Appendix A, pp. 18–19).

### 2.3. Data Extraction and Analysis

Data were extracted from the included studies using a pre-defined data-extraction form. The data extracted for laboratory studies included study setting, study design, assay, method type, study treatment, and main outcome. The data extracted for clinical studies included study setting, study design, study participants, intervention, outcome measures, treatment outcome, and adverse events (AEs). The distinctions between case series and cohort studies were further clarified by consulting systematic reviews conducted in this area [45,46]. All comparisons are narratively described and presented in Tables.

### 2.4. Methodological Quality Assessments

Given the lack of validated tools for quality assessment of pre-clinical studies, the toxicological data reliability assessment (ToxRTool, validated for evaluating the reliability of toxicological pre-clinical studies) [47], was used in pre-clinical studies. The tool has two parts, one for in vitro (18 criteria) and another for in vivo (21 criteria) studies, and each question in both cases was scored as 1 (criterion met) or 0 (criterion not met). Studies were considered reliable without restrictions (15–18 for in vitro and 18–21 for in vivo), reliable with restrictions (11–14 for in vitro and 13–17 for in vivo), and not reliable (<11 for in vitro and <13 for in vivo) [48]. Also, studies scoring “0” for one of the critical questions (six for in vitro and seven questions for in vivo) were considered not reliable. The methodological quality of the RCTs was assessed using the Jadad scale [49], a validated five-point tool. The trials were scored on a scale of 0 (low quality) to 5 (high quality) based on the reports of randomisation, blinding, withdrawals, and dropouts. Trials scoring ≥3 are considered to have high methodological quality [50]. We used the Joanna Briggs Institute (JBI) tools [51] to assess the methodological quality of non-randomised controlled trials (non-RCTs), including quasi-experimental (0–9 scale), cohort (0–11 scale), case series (0–10 scale), and case (0–8 scale) studies. Each question was given a score of 1 for “Yes”, 0 for “No” while no scoring was given for “unclear” responses. As such, studies scoring ≥ 7, 4–6, and <4 were considered to have high, medium, and low methodological qualities, respectively. All the assessments were independently performed by two authors (S.A.B. and W.T.) and disagreements were resolved through discussion. The detailed criteria used to determine each methodological quality were listed in the Appendix A (pp. 7–16).

## 3. Results

### 3.1. Study Selection

The combined search identified a total of 497 records (Figure 3). After removal of duplicates (*n* = 200) and irrelevant records (*n* = 221), 76 records were eligible for full-text screening. Of these, 59 studies met the inclusion criteria and were included in this systematic review. Also, 12 new eligible records identified during a complementary search were included in this systematic review, making the included studies 71.

### 3.2. Study Characteristics

The reviewed studies were conducted in Europe (*n* = 23) [52,53,54,55,56,57,58,59,60,61,62,63,64,65,66,67,68,69,70,71,72,73], Asia (*n* = 21) [74,75,76,77,78,79,80,81,82,83,84,85,86,87,88,89,90,91,92,93,94], North America (*n* = 15) [95,96,97,98,99,100,101,102,103,104,105,106,107,108,109], Australia and New Zealand (*n* = 11) [110,111,112,113,114,115,116,117,118,119,120], and Africa (*n* = 2) [31,121]. Except the four studies published in Mandarin [87,88,90,94], all studies were published in English. Most (*n* = 41) of the included studies were clinical studies involving 2456 participants, with RCT (*n* = 17) [56,62,63,72,74,77,85,86,88,92,93,107,108,110,111,120,121] or non-RCT (*n* = 24) type study designs [52,58,59,61,69,73,75,76,78,79,82,87,89,90,91,94,98,99,101,102,106,109,113,117]. While 25 of them were solely laboratory-based studies with in vitro (*n* = 24) [31,53,54,55,57,60,64,65,66,67,68,70,71,80,83,84,95,100,104,112,114,115,116,119] and in vivo [96] designs. Whereas the remaining five used a mixed in vitro/clinical [81,97,103,118] and in vivo/clinical [105] approaches. Mites, lice, and fleas were the ectoparasites studied in the included studies, with *Demodex* mite being the most widely investigated ectoparasite. We did not identify studies exploring TTO against bed bugs, chigger mites (red bugs), or sand fleas. The main characteristics of the included studies are summarised in Table 1, Table 2, Table 3, Table 4, Table 5, Table 6, Table 7 and Table 8.

### 3.3. Qualitative Syntheses

Acaricidal effect of TTO and its components against mites.

Fifty-five studies targeted mites, which include *Demodex* mites (*n* = 44) [52,56,58,59,61,62,63,64,69,70,71,72,73,74,75,76,77,78,79,81,82,85,87,88,89,90,91,92,93,94,97,98,99,100,101,102,107,108,109,112,114,117,120,121], scabies mites (*n* = 5) [57,86,113,118,119], and house dust mites (*n* = 6) (Table 1, Table 2, Table 3, Table 4 and Table 5) [60,65,68,80,83,116]. Of these, 14 [57,60,64,65,68,70,71,80,83,100,112,114,116,119] were in vitro studies, while three studies [81,97,118] followed a mixed in vitro/clinical approach, and 38 were clinical (interventional) studies (15 RCTs [56,62,63,72,74,77,85,86,88,92,93,107,108,120,121] and 23 non-RCTs [52,58,59,61,69,73,75,76,78,79,82,87,89,90,91,94,98,99,101,102,109,113,117]).

### 3.4. Demodex Mites

Among 44 studies involving *Demodex* mites, six were in vitro studies [64,70,71,100,112,114], two studies followed a mixed in vitro/clinical approach [81,97], and 36 were clinical studies (14 RCTs [56,62,63,72,74,77,85,88,92,93,107,108,120,121] and 22 non-RCTs [52,58,59,61,69,73,75,76,78,79,82,87,89,90,91,94,98,99,101,102,109,117]) (Table 1 and Table 2).

The in vitro studies (*n* = 8) investigating the acaricidal activity of TTO and its components against *Demodex* mites are presented in Table 1. The tested interventions were TTO (2.5–100%) [64,71,97,114], T4O (terpinen-4-ol, 4%) [100], TTO (2–100%) with T4O (0.5–100%) [70,112], and T4O (4–100%) with γ-terpinene (25–100%), α -terpinene (10–100%), α-terpineol (10–100%), and 1,8-cineole (10–100%) [81]. The outcome variable evaluated in these studies was mite survival time (MST), with studies reporting either the mean [64,70,71,81,97,100] or the median MTS [112,114]. The studies demonstrated that TTO (100%) and T4O (100%) were effective in killing the mites within 3–5 min (mean) [64,71,81,97,100] and 9–10 min (median) [112,114] of their applications. However, the mean MST significantly increased as the concentration of TTO decreased, with a mean MST of 7–15 min for TTO (50%), 13–35 min for TTO (25%), and 22–150 min for TTO (10%) (Table 1) [71,97]. Unlike TTO, T4O dilution did not result in a substantial change in mean MST up to 10% (5 min for 50%, 8 min for 25%, 9 min for 10%, and 40 min for 4% T4O) [81]. Other TTO components such as α-terpineol (100%), γ-terpinene (100%), α -terpinene (100%), and 1,8-cineole (100%) demonstrated MST values of 4–14 min [81]. In sum, TTO (≥10%) and T4O (≥10%) demonstrated promising in vitro activities against *Demodex* mites.

The 38 clinical and mixed approach studies involved 2140 *Demodex* infected participants with 971 in RCT (*n* = 14) and 1169 in non-RCT (*n* = 24) studies (Table 2). Most clinical studies (*n* = 23) [52,56,58,61,62,63,72,73,75,76,77,81,85,88,90,92,94,101,107,108,109,120,121] involved blepharitis patients and the remaining (*n* = 15) targeted patients with blepharitis and meibomian gland dysfunction [79], cylindrical dandruff [69,97], external ocular diseases [117], meibomian gland dysfunction [59,87], ocular demodicosis [78,89,98,99], blepharoconjunctivitis [91,102], recurrent chalazion [82], rosacea [74], and dry eye symptoms [93].

The RCTs were controlled with either placebo [62,74,77,107,108], active comparators [56,63,72,85,88,92,93,121], or ‘no treatment’ [120]. TTO (3–100%) and T4O (2.5–4%) were explored as test interventions in nine [56,62,63,72,74,77,88,93,120] and three RCTs [107,108,121], respectively, while two studies [85,92] used TTO (5% and *no concentration reported for the other study*) as a control intervention. Nine RCTs explored either TTO (5–50%) alone [62,63,77,85,88,92] or T4O (2.5%) alone [107,108,121], while the rest [56,72,74,93,120] tested TTO (3–100%) in combination with other active agents. The combination interventions were TTO (*no concentration reported*) with coconut oil [120], TTO (7.5%) with chamomile oil (*no concentration reported*) [72], TTO (3%) with calendula oil plus borage oil [56], TTO (100%) with permethrin [74], and TTO (5%) with artificial tears and topical steroid [93]. The reported outcome variables included *Demodex* mite count (DMC), *Demodex* mite density, *Demodex* eradication rate (DER), improvement in ocular symptoms (i.e., using ocular surface disease index (OSDI) and other scoring scales), and occurrence of AEs. Except for one study assessing only improvement of the ocular symptoms using sterile wipe containing 2.5% T4O [121], all RCTs reported on *Demodex* mites, of which, one study [74] reported *Demodex* mite density, three studies on DMC [93,107,108], one study [56] on DER, and eight studies [62,63,72,77,85,88,92,120] on both DMC and DER. Among the RCTs reported on *Demodex* mites, all except one [74], also reported on post-treatment ocular symptom improvements. As such, all the studies evaluating DMC demonstrated significant *Demodex* count reduction after treatment (Table 2). TTO (3–50%) also demonstrated a DER of 21–96% in nine studies, with the highest DER for TTO (7.5%) and chamomile oil swab (96%) followed by TTO (5%) ointment (75%), Dr Organic Tea Tree Face Wash™ (containing 38% T4O) (50%), and TTO plus coconut oil sterile wipe (50%) [56,62,63,77,85,88,120]. A significant improvement (*p* < 0.05) was also reported in all RCTs evaluating ocular symptoms. Of 14 studies, 10 [56,62,72,74,77,92,107,108,120,121] assessed the AEs of TTO and its components, reporting either no AEs [56,62,72,77,107,108,120] or mild skin and ocular irritations [74,92,121]. Of note, no AEs were reported in RCTs [56,62,77,120] investigating ≤10% TTO formulated in gel, eyelash shampoo, and eyelid wipes.

Most non-RCTs were cohort studies (*n* = 9) [52,61,73,75,76,78,97,99,117], while the remainder (*n* = 15) were case series [69,82,91,98,101,102], quasi-experimental studies [79,87,90,94], and case studies [58,59,81,89,109]. Only four of the studies [79,87,94,97] included a controlled group. Sixteen [52,58,59,61,75,76,79,82,89,90,97,98,99,101,102,117] explored TTO (0.02–50%) alone, other five studies tested T4O (0.1% [69], 2.5% [73], and *no concentration reported* for three studies [78,81,109]) alone, whereas, the remaining three studies tested combinations of TTO (*no concentration reported*) with flurometholone (anti-inflammatory agent) [87], TTO (*no concertation reported*) swab with meibomian glands compression massage [94], and TTO (50%) with oral ivermectin (200 µg/kg, 1 week apart, antiparasitic agent) [91]. The assessed outcome variables included DMC, DER, improvement in ocular symptoms (i.e., using OSDI or other scoring scales), and occurrence of AEs. Except for three studies reporting on only improvement in symptoms [79,91,117] and one study on cure rate [82], others (*n* = 20) reported either DMC [59,73,76,78,87,90,101,102], DER [52,58,69,94], or both variables [61,75,81,89,97,98,99,109]. In addition, except four studies [52,82,89,97], other clinical studies (*n* = 20) assessed ocular symptom improvement as an outcome variable. All studies evaluating DMC reported a significant reduction in *Demodex* count following the treatments (Table 2). Also, 10 studies testing TTO (10–50%) and T4O (0.1–2.5%) [52,58,69,75,81,89,94,97,98,109] demonstrated a DER of 72.2–100% while one study [61] reported DERs of 0 and 6%, for TTO (5%) ointment and TTO (0.02%) cleansing foam, respectively, and another [99] with DER of 45% for TTO (5%) ointment. All studies reported improvement in ocular symptoms following the interventions. Nine studies [52,59,73,79,82,87,97,98,99] assessed AEs associated with TTO and/or its components. Of these, five studies evaluating TTO (10%) eyelash shampoo plus TTO (4%) eyelid gel [52], TTO wipes (*no concentration reported*) [59,87], TTO (50%) eyelid scrub plus 0.5 mL TTO (*no concentration reported*) eyelash shampoo [82], and T4O (2.5%) eyelid wipes [73] reported no AEs; whereas, minor irritations were reported in four studies investigating TTO (0.02%) eyelid scrub foam [79], TTO (50%) eyelid scrub plus 0.5 mL TTO (*no concentration reported*) eyelash shampoo [97,98], and TTO (5%) eyelid scrub ointment [99].

**Table 1 pharmaceutics-14-01587-t001:** Descriptive characteristics of included laboratory *Demodex* studies (*n* = 8).

Study Setting	Study Design	Method/Assay	Intervention	Outcome Measure(s)	Treatment Outcome(s)	Quality Score
Bulut and Tanriverdi, 2021 [70], Turkey	In vitro (*n* = 4.8 (mean) *Demodex* (no report on species type) randomly assigned to each group)	In vitro killing assay: direct application of test solutions onto epilated eyelashes with mites placed on the glass slides and microscopic examination of their non-viability for 360 min	*n* = mean number of 4.8 mites in each group TTO (2 %, Osto^®^) solution (*n* = 5) TTO (7.5 %, Blefaritto^®^) solution Terpinen-4-ol (T4O, 0.5%, Blefastop plus^®^) wipe Saline solution (Control)	Mite survival time (MST): from treatment to non-viability (absence of limb and body movement during an observation period of 1 min)	MST (Mean ± SD): 95.9 ± 25.2 min for TTO (2%) vs. 67.1 ± 21.8 for TTO (7.5%) vs. 27.3 ± 6.0 for T4O (0.5%) vs. 323.5 ± 21.1 for Saline (*p* < 0.001) MST (Mean ± SD): T4O (0.5%) vs. TTO (2%) vs. TTO (7.5%) (*p* < 0.001); TTO (7.5%) vs. TTO (2%) (*p* < 0.001)	17 (Reliable without restriction)
Yurekli and Botsali, 2021 [71], Turkey	In vitro (*n* = 35 *D. folliculorum* randomly assigned to each group)	In vitro killing assay: direct application of test solutions onto diagnostic Standardized Skin Surface Biopsy samples with mites placed on glass slides and microscopic examination of their non-viability for 240 min	TTO (2.5%) TTO (5%) TTO (10%) (*n* = 5) TTO (25%) (*n* = 5) TTO (50%) (*n* = 11) TTO (100%) (*n* = 21) solutions Permethrin (5%) solution (positive control) Immersion oil (negative control)	MST: from treatment to non-viability (absence of body and leg movements during an observation period of 1 min)	MST (Mean ± SD): 54.0 ± 6.1 min for TTO (2.5%) vs. 39.0 ± 3.9 for TTO (5%) vs. 22.0 ± 2.5 for TTO (10%) vs. 13.0 ± 2.5 for TTO (25%) vs. 7.8 ± 0.6 for TTO (50%) vs. 3.3 ± 1.3 for TTO (100%) (*p* < 0.001) vs. 12.5 ± 1.9 for Permethrin 5% vs. 196.0 ± 23.6 for Immersion oil MST (Mean ± SD): 13.0 ± 2.5 for TTO (25%) vs. 12.5 ± 1.9 for Permethrin 5% (*p* = 0.628) (no *p*-value is reported for TTO solutions vs. negative control)	17 (Reliable without restriction)
Cheung et al., 2018 [112], New Zealand	In vitro (*n* = 93 *Demodex* (no report on species type) randomly assigned to each group)	In vitro killing assay: direct application of test solutions onto epilated eyelashes with mites placed on the glass slides and microscopic examination of their non-viability for 300 min	TTO (100%) solution (*n* = 10) TTO (50%) solution (*n* = 10) Terpinen-4-ol (T4O, 100%) solution (*n* = 11) Linalool (100%) solution (*n* = 10) T4O (Cliradex^®^, 4 mg/mL) towelette cleanser (*n* = 10) T4O (Oust^™^ Demodex^®^, 0.29 mg/mL) cleanser (*n* = 11) T4O (Blephadex^™^, 0.03 mg/mL) eyelid foam (*n* = 10) T4O (0.02 mg/mL) and linalool (76%) (TheraTears^®^ SteriLid^®^) eyelid cleanser (*n* = 11) No treatment (*n* = 10)	Mite survival time (MST): from treatment to non-viability (absence of limb and body movement over two consecutive observations periods)	MST (Median [range]): 10 (7–24) mins for TTO (100%) vs. 28 (24–75) for TTO (50%) vs. 12 (5–18) for T4O (100%) vs. 7 (5–21) for Linalool vs. 37.5 (15–240) for Cliradex^®^ vs. 90 (30–150) for Oust^™^ Demodex^®^ vs. 60 (15–240) for Blephadex^™^ vs. 70 (30–145) for TheraTears^®^ SteriLid^®^ vs. ≥ 300 min for No treatment groups (*p* < 0.0001)	17 (Reliable without restriction)
Frame et al., 2018 [114], New Zealand	In vitro (*n* = 52 *Demodex* (no report on species type) randomly assigned to each group)	In vitro killing assay: direct application of TTO solutions onto epilated eyelashes with mites placed on glass slides or placing the mites onto slides smeared with the honey and microscopic examination of their non-viability for 240 min	TTO (100%) solution (*n* = 10) TTO (50%) solution (*n* = 12) Cyclodextrin- complexed manuka honey MGO^™^ (CyCMH, *n* = 12) Uncomplexed manuka honey MGO^™^ (UCMH, *n* = 10) No treatment (*n* = 8)	MST: from treatment to non-viability (absence of limb and body movement)	MST (Median (range)): 9 (6–10) mins for TTO (100%) vs. 121 (8–190) for TTO (50%) vs. 141 (34–185) for CyCMH vs. 190 (190–censored) for UCMH vs. ≥ 250 min for No treatment groups (*p* < 0.001)	17 (Reliable without restriction)
Gao et al., 2005 [97], USA	In vitro (*n* = 116 *D. folliculorum* mites randomly assigned to each group)	In vitro killing assay: direct application of test solutions onto epilated eyelashes with mites placed on glass slides and microscopic examination of their non-viability for 150 min	TTO (100%) (*n* = 21); TTO (50%) (*n* = 11); TTO (25%) (*n* = 5) TTO (10%) (*n* = 5) solutions; Baby shampoo (50%) (BS, *n* = 9); Mineral oil (MO, *n* = 5); Povidone-iodine (10%) (PI, *n* = 4); Alcohol (100%) (Alc, *n* = 7); Alcohol (75%) (Alc, *n* = 8); Caraway oil (100%) (CWO, *n* = 16); Dill weed oil (100%) (DWO, *n* = 5); and Pilocarpine (Pilo, *n* = 3)	MST: from treatment to non-viability (absence of limb and body movement)	MST (Mean ± SD): 3.7 ± 0.8mins for TTO (100%) vs. 14.8 ± 9.5 for TTO (50%) vs. 34.7 ± 4.3 for TTO (25%) vs. 150 (no SD) for TTO (10%) vs. 150 (no SD) for BS vs. 150 (no SD) for MO vs. 150 (no SD) for PI vs. 39 ± 1.2 for 100% Alc vs. 150 (no SD) for 75% Alc vs. 4.4 ± 2.5 CWO vs. 14 ± 8.3 for DWO vs. 150 (no SD) for Pilo (no *p*-value is reported) TTO: 3.7 ± 0.8mins for TTO (100%) vs. 14.8 ± 9.5 for TTO (50%) vs. 34.7 ± 4.3 for TTO (25%) vs. 150 (no SD) for TTO (10%) (*p* < 0.01)	16 (Reliable without restriction)
Kabat 2019 [100], USA	In vitro (*n* = 35 *D. folliculorum* randomly assigned to each group)	In vitro killing assay: immersion of epilated eyelashes with mites placed on glass slides with test solutions and microscopic examination of their non-viability for 90 min	T4O (4%) solution (*n* = 12) Hypochlorous acid (0.01%) solution (HOCl, *n* = 14) Mineral oil (100%) (MO, *n* = 9)	MST or kill time: from treatment to non-viability (absence of limb and body movement)	MST (Mean ± SD): T4O: 40 ± 0.0 min for T4O vs. 87.9 ± 4.2 for HOCl (*p* = 0.0005) HOCl: 87.9 ± 4.2mins for HOCl vs. 90 ± 0.0 for MO (*p* = 0.25)	18 (Reliable without restriction)
Oseka and Sedzikowska, 2014 [64], Poland	In vitro (*n* = not reported, no report on species type)	In vitro killing assay: immersion of mites in test solutions placed on glass slides and microscopic examination of their non-viability for about 6 days	TTO (50%) solution Sage oil (100%) solution Peppermint oil (100%) solution Aloe oil (100%) solution Seabuckthorn oil (100%) solution Physiological saline (control)	MST: from treatment to non-viability (absence of limb and body movement)	MST (Mean): 7 min for TTO (50%) vs. 7 min for Sage oil vs. 11 min for Peppermint oil vs. 9 h for Aloe vs. 3 days for Seabuckthorn vs. 82 h for Control (no *p*-value is reported)	4 (Not assignable)
Tighe et al., 2013 [81], China	In vitro (*n* = 292, no species type is reported)	In vitro killing assay: immersion of epilated eyelashes with mites placed on glass slides with test solutions and microscopic examination of their non-viability for 150 min	*n* = 6 for each group T4O: 100%; 50% 25%, and 10% solutions γ-Terpinene: 100%; 50% and 25% α -Terpinene:100%; 50% 25%, and 10% α-Terpineol: 100%; 50% 25%, and 10% 1,8-Cineole: 100%; 50% 25%, and 10% Mineral oil (100%) control (NB: only the top five major components are considered here)	MST: from treatment to non-viability (absence of movement of legs)	MST (Mean ± SD): T4O: 3.6 ± 1.1 min for 100% vs. 4.5 ± 1.0 for 50% vs. 8.3 ± 3.1 for 25% vs. 12.3 ± 8.8 for 10% T4O; γ-Terpinene: 8.3 ± 6.2 for 100% vs. 75.9 ± 29.8 for 50% vs. > 150 for 25%; α -Terpinene:13.6 ± 4.4 min for 100% vs. 21.0 ± 2.2 for 50% vs. 61.6 ± 11.6 for 25% vs. > 150 for 10%; α-Terpineol: 3.8 ± 0.8 for 100% vs. 12.5 ± 2.9 for 50% vs. 22.8 ± 3.9 for 25% vs. 43.4 ± 4.3 for 10%; 1,8-Cineole: 13.5 ± 2.0 for 100% vs. 18.8 ± 4.1 for 50% vs. 23.5 ± 3.9 for 25% vs. 44.4 ± 7.2 for 10% vs. no effect for MO (no *p*-value is reported for each comparison)	17 (Reliable without restriction)

**Table 2 pharmaceutics-14-01587-t002:** Descriptive characteristics of included interventional and observational *Demodex* studies (*n* = 38).

Study Setting	Study Design	Study Participant	Intervention Description	Outcome Measure(s)	Treatment Outcome(s)	Quality Score
RCTs (*n* = 14)						
Ebneyamin et al., 2019 [74], Iran	Randomized double-blind, placebo-controlled trial	Rosacea patients with *Demodex* (age = not reported, *n* = 47)	Test (*n* = 35 right side faces): Received permethrin (2.5%) with TTO (100%) gel, applied on the skin BID (Twice daily) for 12 weeks Control (*n* = 35 left side faces): Received placebo gel	Demodex mite density (DMD/cm^2^) after 12 weeks AEs occurrence	DMD (Mean): 528.8 (BL (baseline):1346) in Test vs. 650.9 (BL:1407.1) in Control (*p* = 0.001) AEs: No allergic reactions and no major AEs observed but skin dryness (*n* = 21, 60.0% moderate and 37.1% mild), burning and stinging (*n* = 7, 20%), erosion (*n* = 7, 20%) and erythema (*n* = 3, 8.6%)	5 (High)
Epstein et al., 2020 [108], USA	Randomized double-blind, placebo-controlled trial	Blepharitis patients with *Demodex* (Mean age: 71.0 ± 6 5.8 years in Test and 75.6 ± 5.0 years in Control groups, *n* = 50)	Test (*n* = 26): Received microblepharoexfoliation (MBE, one application at baseline) plus Cliradex^®^ eyelid scrubs (T4O, no concentration reported) applied BID for 1 month. Control (*n* = 24): Received MBE (one application at baseline) plus sham scrubs (no medication, content not reported) applied BID for 1 month. After 1 month, both test and control groups received MBE (one time application) plus Cliradex^®^ eyelid scrubs BID for 1 month	Demodex mite count (DMC, per four epilated lashes) after 1 month DMC after 2 months Ocular Surface Disease Index (OSDI) score after 1 month:1–100 scale OSDI score after 2 months:1–100 scale AEs occurrence	DMC (Mean ± SD) after 1 month: 3.6 ± 1.5 (BL:4.7 ± 1.5) in Test group (*p* = 0.266) vs. 3.0 ± 1.0 (BL:5.1 ± 1.4) in Control group (*p* = 0.015) DMC (Mean ± SD) after 2 months: 2.6 ± 1.2 (BL:4.7 ± 1.5) in Test group (*p* = 0.026) vs. 2.5 ± 0.9 (BL:5.1 ± 1.4) in Control group (*p* = 0.005) OSDI score (Mean ± SD) after 1 month: 15.1 ± 8.9 (BL: 19.1 ± 8.5) in Test group (*p* = 0.505) vs. 17.2 ± 8.5 (BL: 16.9 ± 7.9) in Control group (*p* = 0.962) OSDI score (Mean ± SD) after 2 months: 16.6 ± 7.9 (BL: 19.1 ± 8.5) in Test group (*p* = 0.660) vs. 7.7 ± 5.4 (BL: 16.9 ± 7.9) in Control group (*p* = 0.074) AEs: Both treatments were well tolerated and burning, or irritation symptoms reported by few patients (no specific number reported) dissipating in minutes or less.	5 (High)
Ergun et al., 2020 [56], Turkey	Randomized double-blind, placebo-controlled trial	Blepharitis patients with *Demodex* (Mean age: 48.80 ± 13.22 years in Test and 53.16 ± 9.59 in Control groups, *n* = 49)	Test (*n* = 25): Received advanced cleansing gel formulation containing 3% (*w*/*w*) TTO plus < 5% (*w*/*w*) calendula oil, borage oil, vitamin E, vitamin B5 BID for 1 month Control (*n* = 24): Received basic cleansing gel formulation containing 3% (*w*/*w*) TTO BID for 1 month	Demodex Eradication rate (DER) after 1 month Ocular Surface Disease Index (OSDI) score after 1 month AEs occurrence	DER (%): 20.6% (BL:54.2%) in advanced gel (*p* = 0.004) vs. 27.8% (BL:42.0%) in Basic cleansing gel (*p* = 0.302) OSDI score (Mean ± SD): 24.0 ± 16.1 (BL:44.3 ± 22.5) in Advanced gel (*p* = 0.001) vs. 18.7 ± 15.0 (BL:36.5 ± 17.8) in Basic cleansing gel (*p* = 0.001) AEs: No AEs were observed in both groups	4 (High)
Karakurt and Zeytun, 2018 [62], Turkey	Randomised single-blinded controlled trial	Blepharitis patients with *Demodex* (Mean age: 56.5 ± 14.1, *n* = 135)	Test (*n* = 75): Received TTO (7.5%) eyelash shampoo applied BID for 4 weeks Control (*n* = 60): Received TTO-free eyelash shampoo applied BID for 4 weeks	Demodex mite count (DMC) after 1 month DER Ocular symptoms (itching, burning, foreign body sensation, redness, and cylindrical dandruff) score: 0–3 AEs occurrence	DMC (Mean): 0 (BL: 6.3) in 36% (27/75) (*p* < 0.001) and 4.2 (BL:12.5 per eyelash) in 64% (48/75) of patients (*p* < 0.001) in TTO group vs. 0 (BL:2.0) in 11.7% (7/60) (*p* = 0.017) and 7.9 (BL: 12.0 per eyelash) in 89.3% (53/60) of patients (*p* = 0.024) in Control group DER (%): 36% (27/75) in TTO group vs. 11.7% (7/60) in Control group Ocular symptoms score (Mean): Decreased in Test (*p* < 0.001) vs. Remained the same in Control group (*p* > 0.05) AEs: No irritation or other side effect complaints for both groups	2 (Low)
Koo et al., 2012 [77], South Korea	Randomized controlled trial	Blepharitis patients with *Demodex* (Mean age: 55.7 ± 12.4 years, *n* = 281)	Test (*n* = 141): Received TTO (50%) lid scrub weekly followed by TTO (10%) lid scrub daily applied for 1 month Control (*n* = 140): Received eyelid scrub with saline	DMC (per eight epilated lashes) after 1 month DER after 1 month OSDI score after 1 month Patient compliance (for TTO group): good (> 10 times scrubbing/week); moderate 5–9 times/week) and poor (< 5 times scrubbing/week) AEs occurrence	DMC (mean ± SD): 3.2 ± 2.3 (BL:4.0 ± 2.5) in TTO group (*p* = 0.001) vs. 4.2 ± 2.5 (BL:4.3 ± 2.7) in Control group (*p* = 0.27) (*p* = 0.004) DER (%): 23.6% (25/106) in TTO group vs. 7% (4/54) in Control group OSDI score (mean ± SD): 24.1 ± 11.9 (BL:34.5 ± 10.7) in TTO group (*p* = 0.004) vs. 27.5 ± 12.8 (BL:35.3 ± 11.6) in Control group (*p* = 0.04) Patient compliance: 37.7% (40/106) with good vs. 34% (36/106) with moderate vs. 28.3% (30/106) with poor compliance (no report on patient compliance for control) AEs: 4.7% (5/106) reported ocular irritation but disappeared following patient’s education on the proper scrubbing method	2 (Low)
Liu and Gong, 2021 [92], China	Randomized controlled trial	Blepharitis patients with *Demodex* (Mean age: 46.2 ± 13.0years, *n* = 52)	Test (*n* = 27): Received okra eyelid patch (no concertation reported) applied every night for 3 months Control (*n* = 25): Received TTO eye care patch (no concertation reported) applied every night for 3 months	DMC (per four epilated lashes) after 3 months DER after 3 months OSDI score (0–100) after 3 months AEs occurrence	DMC (mean ± SD): 1.3 ± 1.4 (BL:10.2 ± 4.5) in Test group vs. 1.9 ± 0.2 (BL: 11.2 ± 5.9) in Control group (*p* = 0.716) DER (%): 40.74% (11/27,) in Test group vs. 48% (12/25) in Control group OSDI score (mean ± SD): 23.7 ± 10.7 (BL: 40.5 ± 10.9) in Test group vs. 18.4 ± 3.3 (BL: 35.9 ± 12.8) in Control group (*p* = 0.873) AEs: 3.7% (1/27, ocular pruritus and discomfort) in Test group vs. 16% (4/25, slight to moderate irritation with conjunctival congestion) in Control group	3 (High)
Mergen et al., 2021 [72], Turkey	Randomised double-blind, active comparator-controlled trial	Seborrheic blepharitis patients with *Demodex* (Mean age: 28.4 ± 65.2years in Test and 31.8 ± 61.1years in Control groups, *n* = 52)	Test (*n* = 26): Received TTO (7.5%) and chamomile oil (no concentration reported) swabs applied BID for 2 months and followed by a month of treatment withdrawal period Control (*n* = 26): Received Johnson’s Baby Shampoo (BS) applied BID for 2 months followed by a month of treatment withdrawal period	DMC (per four epilated lashes) after 2 months DER after 2 months OSDI score after 2 months Blepharitis Symptom measure (BLISS) score after 2 months AEs occurrence	DMC (mean ± SD): 0.0 ± 0.1 (BL: 1.5 ± 1.1) (*p* < 0.001) in Test group vs. 0.0 ± 0.1 (BL:1.2 ± 1.0) (*p* < 0.001) in Control group (*p* = 0.930) DER (%): 95.5% (21/22) in Test group vs. 95.7% (22/23) in Control group (no *p* value reported) OSDI score (mean ± SD): 7.7 ± 7.2 (BL: 16.5 ± 16.0) (*p* < 0.001) in Test group vs. 12.3 ± 11.0 (BL: 13.0 ± 8.8) (*p* = 0.143) in Control group (*p* = 0.186) BLISS score: (mean ± SD): 1.1 ± 2.8 (BL: 10.0 ± 4.0) (*p* < 0.001) in Test group vs. 6.6 ± 6.7 (BL: 9.6 ± 4.4) (*p* = 0.01) in Control group (*p* < 0.001) AEs: No patients reported AEs in both groups	5 (High)
Messaoud et al., 2019 [121], Tunisia	Randomized open level-controlled trial	Blepharitis patients with *Demodex* (Mean age: 52.0 ± 16.2 in Test group I and 56.5 ± 15.1 in Test group II, *n* = 48)	Test I (*n* = 24): Received T4O (2.5%) plus hyaluronic acid (0.2%, moisturizing agent) sterile wipe (Blephademodex^®^) once daily for 29 days Test II (*n* = 24): Received T4O (2.5%) plus hyaluronic acid (0.2%, moisturizing agent) sterile wipe (Blephademodex^®^) BID for 29 days Control: None	Reduction in overall ocular discomfort on Day 29 (0–10 points) Improvement in ocular symptoms score (itching, burning/stinging and foreign body sensation) on Day 29 (0–5 points) Patient satisfaction (Day 29) AEs occurrence (Day 29)	Reduction in overall ocular discomfort (mean ± SD): 1.1 ± 1.0 (BL: 6.4 ± 1.4, *p* < 0.0001) Test group I vs. 0.2 ± 0.8 (BL: 7.0 ± 1.5, *p* < 0.0001) in Test group II (*p* = 0.718) Improvement in overall ocular symptoms: satisfactory or very satisfactory in 95.7% in Test group I vs. 100% in Test group II Patient satisfaction: 100% for both groups AEs: 1/24 (moderate burning sensation after application which resolved after 3s) in Test group I vs. 2/24 (visual acuity) in Test group II	2 (Low)
Mohammadpour et al., 2020 [93], Iran	Randomised triple-blinded controlled trial	Patients with dry-eye symptoms after cataract surgery (Mean age: 66.4 ± 8.8 years, *n* = 62, of these *n* = 43 with *Demodex*: *n* = 23 in the Test and *n* = 18 in the Control groups)	Test (*n* = 33): Received eyesol shampoos with TTO (5%), artificial tears, and topical steroid TID for 1 month Control (*n* = 29): Received eyesol shampoos without TTO, artificial tears and betamethasone (1%) drops TID for 1 month	DMC (per four epilated lashes) after 1 months OSDI score after 1 months	DMC (mean ± SD): 0.9 ± 2.3 (BL: 2.4 ± 2.9) (*p* < 0.001) in Test group vs. 2.7 ± 3.3 (BL:2.7 ± 3.9) (*p* = 0.916) in Control group (*p* = 0.024) OSDI score (mean ± SD): 21.9 ± 19.1 (BL: 42.5 ± 25.1) (*p* < 0.001) in Test group vs. 31.5 ± 22.6 (BL: 41.1 ± 26.4) (*p* < 0.05) in Control group (*p* < 0.05)	4 (High)
Murphy et al., 2018 [63], Ireland	Randomised controlled trial	Blepharitis patients with *Demodex* (Mean age: 49.6 ± 17.1 years in TTFW, 49.6 ± 16.9 in OLSP and 49.86 ± 19.7 in BlephEx^™^ groups, *n* = 69, *n* = 17 participants with no Demodex mites)	Test (*n* = 22): Received TTO containing 38% T4O (Dr Organic Tea Tree Face Wash^™^, TTFW) lid scrub daily (night-time) for 4 weeks Test II (*n* = 24): Received OcuSoft Lid Scrub Plus (OLSP) wipes (Active ingredient: 0.5%1, 2-Octanediol) daily (night-time) for 4 weeks Test III (*n* = 23): Used BlephEx^™^ exfoliation device once at initial visit and received OLSP wipes at home nightly for 4 weeks	DMC after 4 weeks DER after 4 weeks OSDI score after 4 weeks	DMC (median [range]): 1.9 (0–8) (BL:4.9[0–21]) (*p* = 0.001) in TTFW group vs. 1.9(0–7) (BL:3.8[0–11]) (*p* = 0.005) in OLSP group vs. 2.7 (0–9) (BL:6.5[1–25]) (*p* = 0.001) in BlephEx^™^ group (*p* = 0.498) DER (%): 40.9 % (9/22) in TTFW group vs. 45.8% (11/24) in OLSP group vs. 39.1% (9/23) in BlephEx^™^ group OSDI score (mean ± SD): 16.2 ± 15.2 (BL:27.4 ± 16.7) in TTFW group vs. 13.6 ± 17.1 (BL:28.6 ± 23.6) in OLSP group vs. 12.8 ± 12.8 (BL:30.1 ± 19.8) in BlephEx^™^ group (*p* = 0.646)	2 (Low)
Tseng S. (NCT 01647217), 2017 [107], USA	Randomised controlled trial	Chronic blepharitis patients with *Demodex* (Mean age: 48.8 ± 19.1 years, *n* = 17)	Test (*n* = 8): Received T4O (Cliradex^®^) lid scrub (no concentration reported) once or twice per day for 1 month Control (*n* = 9): Received placebo lid scrub once or twice per day for 1 month	DMC after 6 weeks Lid Margin Redness and Bulbar Conjunctival Hyperemia: 0 (none)- 6 (severe) after 6 weeks AEs occurrences	DMC (Mean change ± SD): -3 ± 3.1 in Test group vs. -0.4 ± 3.6 in Control group Lid Margin Redness and Bulbar Conjunctival Hyperemia (Mean change ± SD): -2.3 ± 1.4 in Test group vs. -3.1 ± 1.9 in Control group AEs: 0% (0/8) in Test group vs. 0% (0/9) in Control group (no *p*-value is reported)	NA as this is only trial registry record
Wang et al., 2020 [88], China	Randomised controlled trial	Blepharitis patients with *Demodex* (Mean age: 37 ± 14 years, *n* = 32 with 64 eyes)	Test (*n* = 16, 32 eyes): Received TTO eye patch (concertation not reported) BID combined with daily (night-time) eyelid margin deep cleaning in one eye for 3 months Control (*n* = 16, 32 eyes): Received TTO eye patch (concertation not reported) BID in the other eye for 3 months	DMC after 3 months DER after 3 months OSDI score after 3 months (Only the outcomes with clinical significance are considered for this study)	DMC (median [range]): 1 (0–2) (BL:6 [4–9], [*p* < 0.01]) in Test group vs. 2 (0–2) (BL:6 [5–11] [*p* < 0.01]) in Control group (*p* = 0.022) DER (%): 37.5% (12/32 eyes) in Test group vs. 28.1% (9/32 eyes) in Control group OSDI score (median (range)): 54.5 (27.1–65.0) Pre-treatment vs. 28.1 [16.3–52.7] Post-treatment in both groups (*p* < 0.001)	3 (High)
Wong et al., 2019 [120], Australia	Randomised single blinded (R vs. L eye) controlled pilot trial	Blepharitis patients with *Demodex* (Median age: 63.5 (range 48–76)) years, *n* = 20)	Test (*n* = 20 eyes): Received TTO and coconut oil (Blephadex^™^ concentrations not reported) Eyelid Wipes in one eye once daily for 1 month Control (*n* = 20 eyes): The contralateral eye was left untreated	DMC after 1 month DER (DMC reduction to 0) after 1 month OSDI (1–100) after 1 month AEs occurrence	DMC (Median ± IQR): 0 ± 2 (BL:2 ± 3) in Test vs. 2 ± 4 (BL:3 ± 5) in Control group (*p* = 0.04) DER (%): 50% in Test vs. 29% in Control group OSDI (Median ± IQR): 9 ± 14 (BL:9 ± 15) in Test vs. 9 ± 14 (BL:9 ± 15) in Control group (*p* = 0.15) AEs: No AEs observed and product well tolerated by participants	3 (High)
Zhang et al., 2019 [85], China	Randomized controlled trial	Blepharitis patients with *Demodex* (Mean age: 38.3 ± 12.3 years in IPL and 39.2 ± 11.0 in TTO groups, *n* = 40)	Test (*n* = 20): Received intense pulsed light (Lumenis^®^ M22TM) treatments three times in 3 months Control (*n* = 20): Received TTO (5%) ointment 15 min lid massage daily for 3 months	DMC (per eight epilated lashes) after 3 months DER after 3 months OSDI score after 3 months	DMC (mean ± SD): −13.1 ± 8.5 (BL:13.1 ± 8.5) in Test vs. −11.1 ± 6.9 (BL:12.9 ± 6.5) in Control (*p* = 0.780) DER (%): 100% (20/20) in Test vs. 75% (15/20) in Control OSDI score (mean ± SD): −25.6 ± 31.0 (BL:30.5 ± 30.5) in Test vs. −15.6 ± 27.8 (BL:33.5 ± 29) in Control (*p* < 0.01)	2 (Low)
Non-RCTs (*n* = 24)						
Alver et al., 2017 [52], Turkey	Cohort study	Blepharitis (chronic and treatment-resistant) patients with Demodex (mean age = 54.1 ± 15.4 years, *n* = 39)	Test (*n* = 28): Received TTO (10%) eyelash shampoo with TTO (4%) eyelid gel, both applied on the eyelids BID for 1 month Control: None	DER, % after 1 month OSDI score after 1 month Improvement in symptoms (%) AEs occurrence	DER (%): 82.1% (23/28) (no *p*-value is reported) Improvement in symptoms: 89.2% (25/28) (no *p*-value is reported) OSDI score (Mean ± SD, *n* = 12): 33.0 ± 2.7 (BL:39.6 ± 10.1) (*p* = 0.002) AEs: No patient complained of the TTO use	5 (Medium)
Evren Kemer et al., 2020 [69], Turkey	Case series	Cylindrical dandruff (CD) patients with *Demodex* (Mean age: 52.8 ± 15.8 years, *n* = 30)	Test (*n* = 30): Received eye warm compressed at 43–45 °C for 5 min followed by cleaning eyelids with T4O (0.1%) plus sodium hyaluronate (moisturiser) wipes (Blefastop plus^®^) BID for 2 weeks (first cycle treatment). After 7–10-days washout period, the same treatment repeated (second cycle treatment) Control: None	DER after 2weeks and 1 year OSDI score after first cycle treatment (3 weeks), second cycle treatment (6 weeks) and 1 year Treatment compliance (Only the outcomes with clinical significance are considered for this study)	First cycle: OSDI score (Mean ± SD): 34.3 ± 13.4 (BL: 48.0 ± 19.8) (*p* = 0.001) Second cycle DER (%): 86.7% (27/30) (no *p*-value is reported) OSDI score (Mean ± SD): 40.1 ± 21.1 (BL:48.0 ± 19.8) (*p* = 0.001) After 1 year, DER: 86.7% (27/30) (no *p*-value is reported) OSDI score (Mean ± SD): 41.3 ± 14.6 (BL:48.0 ± 19.8) (*p* = 0.001) Treatment compliance: 86.7% (27/30)	8 (High)
Galea et al., 2014 [58], UK	Case study	A blepharitis patient with *Demodex* (age = 60 years, *n* = 1)	Test (*n* = 1): Received TTO (5%) ointment and tea tree lid scrub (50%) for 3 months Control: None	DER after 3 months Blepharitis improvement	DER (%):100% or complete eradication of the mites Symptom improvement Significant improvement of blepharitis (no *p*-value is reported)	7 (High)
Gao et al., 2005 [97], USA	Cohort study	Cylindrical dandruff (CD) patients with *Demodex* (mean age = 59.9 ± 8.7, *n* = 16)	Test (*n* = 9): Received weekly (three-time application) of TTO (50%) lid scrub at the office plus daily (two times) application of 0.5 mL tea tree shampoo (TTO < 10 %) lid scrub for 1 month and then once daily thereafter at home Control (Conventional treatment, *n* = 7): Received daily lid hygiene with baby shampoo	DMC from epilated lashes with CD after 1 month DER after 1 month AEs occurrence	DMC (Mean ± SD):0 in 7 patients (BL:7.9 ± 4.1) in 4 weeks in Test vs. Never zero in 50 weeks in Control DER (%): 77.8% in Test vs. 0% in Control AEs: TTO (50%) generated irritation in some patients (no data is reported) (no *p*-value is reported)	9 (High)
Gao et al., 2007 [98], USA	Case series	Ocular demodicosis patients with *Demodex* (Mean age: 60.2 ± 11.6 years, *n* = 11)	Test (*n* = 11): Received TTO (50%) office lid scrub weekly and 0.5 mL Tea Tree shampoo lid scrub BID for 1 month Control: None	DMC (per eight lashes) after 1 month DER after 1 month Improvement in symptoms (inflammation) after 1 month AEs occurrence after 1 month	DMC: 5 (BL:120) in all patients and 0 (BL:17 ± 5.2) in 8 patients DER (%): 72.2% (8/11) Symptom improvement: 81. 8% (9/11) patients showed 50–100% improvement in symptoms AEs: TTO (50%) office lid scrub caused mild irritation in 3 and moderate irritation in 6 participants (no *p*-value is reported)	8 (High)
Gao et al., 2012 [99], USA	Cohort study	Ocular demodicosis patients (Mean age: 37.2 ± 15.6 years, *n* = 24)	Test (*n* = 24): Received TTO (5%) ointment lid massage BID for 1 month Control: None	DMC (per eight epilated lashes) after 1 month DER after 1 month Itching grades: Grades 1 (mild), 2 (moderate), and 3 (severe) AEs occurrence	Mean DMC: 0.7 ± 0.8 (BL:4.6 ± 1.8) (*p* < 0.01, *n* = 24) and 0 (*n* = 11 patients) DER (%): 45.8% (11/24) Itching: 66.7% (16/24) no itching while 7 subjects (BL:6) Grade 1 vs. 1 (BL:14) Grade 2 vs. 0 (BL:4) Grade 3 (*p* < 0.01) AEs: Mild ocular irritation in 2 participants	7 (High)
Gunnarsdóttir et al., 2016 [59], Iceland	Case study	Meibomian gland dysfunction (MGD) patients with *Demodex* (Age: 35 and 72 years, *n* = 2)	Test (*n* = 2): Applied Tea Tree wet wipes (TTO concentration not stated) on eyelashes, eyebrows, and face BID for 10 weeks Control: None	DMC (mites/eye) after 10 weeks OSDI after 10 weeks AEs occurrence	DMC: 2–4 mites (BL:8–12 mites per eyes) in both patients Or Reduction in DMC:66.7–75.8% OSDI score: 16.7 (BL:35.4) in both patients AEs: no side effects in both patients (no *p*-value is reported)	8 (High)
Hirsch-Hoffmann et al., 2015 [61], Switzerland	Cohort study	Blepharitis patients with *Demodex* mites (age = not reported, *n* = 96)	Test: Received daily lid hygiene plus TTO (5%) ointment applied once daily (*n* = 6); TTO (0.02%) cleansing foam applied once daily (*n* = 38); metronidazole (MTZ, 2%) ointment applied once daily (*n* = 5); Ivermectin tablets (IVM, 6 mg given po at Days 1 and 14) (*n* = 27); MTZ (500 mg po BID for 10 days) Control: None	DMC (10 epilated lashes) after 2 months DER after 2 months Symptom improvement Treatment preference AEs occurrence	DMC: 13.3 for TTO ointment vs. 12.0 for TTO foam vs. 9.4 for MTZ ointment vs. 12.8 for IVM (oral) vs. 22.0 for MTZ (oral) (no baseline data and *p*-value are reported) DER (%): 0% for TTO ointment vs. 6% for TTO foam vs. 0% for MTZ ointment vs. 6% for IVM (oral) vs. 0% for MTZ (oral) (no *p*-value is reported) Symptom improvement (%):20% for TTO ointment vs. 40.5% for TTO foam vs. 20% for Metronidazole ointment vs. 35% for IVM (oral) vs. 20% for MTZ (oral) Treatment preference: 2/96 (2.1%) for daily lid hygiene vs. 7/96 (7.3%) for TTO ointment vs. 45/96 (46.9%) for TTO foam vs. 5/96 (5.2%) for MTZ ointment vs. 32/96 (33.3%) for oral IVM vs. 5/96 (5.2%) for oral MTZ AEs: no AEs for systemic drugs but AEs not reported for topical treatments	3 (Low)
Huo et al., 2021 [89], China	Case study	Patients with *Phthirus pubis* and *Demodex* co-infestation (Age: 48 years, *n* = 1)	Test: Received TTO (25%) daily lid scrubs and applied for 2 months Control: None	DMC (12 epilated lashes) after 2 months DER after 2 months	DMC: 0 (BL:19 mites) DER (%): 100% (2/2 eyes) (no *p*-value is reported)	7 (High)
Jacobi et al., 2021 [73], Germany	Cohort study	Blepharitis patients with *Demodex* (Mean age: 60.9 ± 18.7 years, *n* = 50)	Test (*n* = 6): Received T4O (2.5%) plus hyaluronic acid (0.2%, moisturiser) eyelid wipes (Blephademodex^®^) every evening for 28 days Control: None	DMC (10 epilated lashes) after 28 days Global discomfort scale (GDS) after 28 days:0 (no)–10 (worst) scale Total ocular symptom score (TOSS): 0 (none)- 4 (all the time) The symptom assessment in dry eye (SANDE) score: very mild–very severe Patient satisfaction after 28 days Treatment compliance after 28 days AEs (tolerability) occurrence after 28 days	Results are for mean changes from 0 to 28 days (only initial treatment phase) DMC (Mean change ± SD): −1.5 ± 1.7 (*p* < 0.0001) GDS (Mean change ± SD): −1.9 ± 1.9 (*p* < 0.0001) TOSS (Mean change ± SD): −18.7 ± 16.2 (*p* < 0.0001) SANDE (Mean change ± SD): −1.9 ± 2.2 (*p* < 0.0001) Patient satisfaction: 66 % (42% satisfied and 24% very satisfied) Treatment compliance: all patients were regarded as compliant AEs: 86% of participants rated the T4O-wipes tolerable and no AEs were reported during the study period	8 (High)
Kheirkhah et al., 2007 [101], USA	Case series	Blepharitis patients with *Demodex* (Mean age: 49.3 ± 17 years, *n* = 6)	Test (*n* = 6): Received TTO (50%) weekly lid scrubs and daily tea tree shampoo lid scrubs applied for 6 weeks Control: None	DMC (per eight lashes) after 6 weeks Improvement of symptoms after 6 weeks	DMC (Mean ± SD): 1 ± 0.9 (BL:6.8 ± 2.8) (*p* = 0.001) Symptom improvement: Dramatic resolution of ocular irritation and inflammatory signs in all participants	7 (High)
Kim et al., 2011 [75], South Korea	Cohort study	Blepharitis patients with *Demodex* (Mean age: 48.3 ± 18.9 years, *n* = 10 and 13 eyes)	Test (*n* = 10): Received TTO (50%) weekly lid scrub and TTO (10%) shampoo lid scrub BID for 1 month Control: None	DMC (per eye) after 1 month DER after 1 month Improvement in symptoms (bulbar conjunctival injection, conjunctival papillary hypertrophy corneal erosions and infiltrations)	DMC (Mean ± SD): 0.2 ± 0.4 (BL:3.8 ± 2.2 per eye) (*p* = 0.001) DER (%): 76.9% (10/13 eyes) Symptom improvement: 53–100% improvements in ocular symptoms in all patients	7 (High)
Kojima et al., 2011 [76], Japan	Cohort study	Blepharitis patients with *Demodex* (Mean age: 62.9 ± 9 years, *n* = 15)	Test (*n* = 15): Received TTO (50%) weekly lid scrubs and tea tree shampoo (10%) daily lid scrubs applied for 6 weeks (*n* = 15 eyes) Control: None	DMC (per epilated lash) after 6 weeks Improvement in symptoms VAS score (itchiness and foreign body sensation) after 6 weeks: 0–100 scale	DMC (Mean ± SD): 0.5 ± 0.5 (BL:4.0 ± 0.5) (*p* < 0.05) Symptoms’ improvement VAS Scores Itchiness VAS Score: 15 ± 5.5 (BL: 92 ± 2.5) Foreign Body Sensation VAS Score: 1.0 ± 1.0 (BL: 96.5 ± 6) Ocular symptoms improved post-treatment (*p* < 0.05)	7 (High)
Liang et al., 2010 [102], USA	Case series	Paediatric blepharoconjunctivitis patients with *Demodex* (Age range:2.5–11 years, *n* = 12)	Test (*n* = 12): Six patients received TTO (50%) eyelid scrubs 3 times/week for 4–6 weeks and the other six (who were not cooperative to the TTO eyelid scrub) received TTO (5%) ointment eyelid massages BID for 4–6 weeks Control: None	DMC (per four epilated lashes) after 6 weeks Improvement in ocular symptoms (surface irritation and reactions, eyelid margin swelling and conjunctival redness)	DMC: Reduced to 0–1 in 4/11 (BL:26 mites for 11 patients) (no DMC report on the *n* = 7 participants) Improvement in ocular symptoms: Dramatic resolution of ocular irritation and inflammation in 2 weeks in all patients (no *p*-value is reported)	6 (Medium)
Liang et al., 2018 [78], China	Cohort study	Ocular demodicosis patients (Mean age: 19.1 ± 7.5 years, *n* = 60 involved and 48 received treatment)	Test (*n* = 48): Received T4O (Cliradex^®^, no concentration reported) lid scrub BID for 3 months Control: None	DMC after 3 months Improvement in ocular symptoms (surface inflammation)	DMC (Mean ± SD): 0.5 ± 0.7 (BL:5.6 ± 3.5) (*p* < 0.001) Improvement in ocular symptoms: Rapidly resolved within 2–3 weeks	8 (High)
Lyu et al., 2021 [90], China	Quasi-experimental	Blepharitis patients with *Demodex* (Mean age: 43.8 ± 11.5 years in OPT group; 44.2 ± 11.1 in TTO group and 44.9 ± 10.7 in OPT + TTO group, *n* = 283)	Test I: Received optimal pulse technology (OPT) 3 times/2 weeks for 6 weeks (*n* = 94) Test II: Received a combination of OPT 3 times/2 weeks and TTO cleansing eye patch daily (night-time) for 6 weeks (*n* = 96) Test III: Received TTO cleansing eye patch daily (night-time) for 6 weeks (*n* = 96)	DMC (per 12 epilated lashes) after 6 weeks Improvement in symptoms (itchiness, burning eyes, and foreign body sensation) after 6 weeks: 0–24 score (Only the outcomes with clinical significance are considered for this study)	DMC (Mean ± SD): 1.3 ± 1.9 (BL:8.3 ± 6.1, *p* < 0.05) in OPT + TTO group vs. 2.4 ± 2.2 (BL:9.3 ± 8.3, *p* < 0.05) in TTO group vs. 5.3 ± 4.1 (BL:9.0 ± 5.5, *p* < 0.05) in OPT group (*p* < 0.01) Improvement in ocular symptoms score (Mean ± SD): 2.8 ± 2.0 (BL:13.4 ± 2.5, *p* < 0.05) in OPT + TTO group vs. 4.8 ± 2.3 (BL:12.8 ± 3.2, *p* < 0.05) in TTO group vs. 4.3 ± 2.3 (BL:13.1 ± 3.3, *p* < 0.05) in OPT group (*p* < 0.01)	9 (High)
Maher 2018 [79], United Arab Emirates	Quasi-experimental	Blepharitis and meibomian gland dysfunction (MGD) patients with *Demodex* (Mean age: 51.5 ± 9.2 years in TTO group and 52.9 ± 9.3 years in Massage group, *n* = 40)	Test (*n* = 20): Received TTO (0.02%) eyelid (Naviblef^™^) scrub foam BID for 1 month Control (*n* = 20): Performed eyelid massage for 5 min QID plus cleansing the lid margins with mild (baby) shampoo QID	Decrease in OSDI score after 1 month Improvement in ocular/lid symptoms (reported by patients) after 1 month AEs occurrence	OSDI score (Mean ± SD): 8.7 ± 4.0 (BL:47.8 ± 8.4) in Test (*p* < 0.001) vs. 30.1 ± 8.9 (BL: 44.3 ± 6.8) (*p* = 0.03) in Control Improvement in ocular symptoms: 100% (20/20) in Test (*p* < 0.001) vs. 25% (5/20) (no *p*-value) in Control AEs: 1 (contact dermatitis) in Test vs. 1 (eye irritation) in Control	9 (High)
Nicholls et al. 2016 [117], Australia	Cohort study	External ocular diseases patients with *Demodex* (Mean age: 62 years, *n* = 333)	Test (*n* = 333): Received TTO (5%) ointment daily (night-time) for 3 months Control: None	Improvement in symptoms (anterior blepharitis, chronic primary conjunctivitis dry eye disease, MG disease and allergic conjunctivitis) after 3 months: 0 (no symptom) –5 (severe) scale	Improvement in symptoms: 91.4% (213/233) some improvement; 10.3% (24/233) complete resolution; 16.8% (40/233) very little problem; 28.9% (67/233) much better; 26.7% (62/233) somewhat better; 8.6% (20/233) just a little better; 8.6% (20/233) no change in the symptoms (no *p*-value is reported)	5 (Medium)
Patel et al. 2020 [91], India	Case series	Blepharokeratoconjunctivitis patients with *Demodex* (Mean age: 19.1 ± 7.5 years, *n* = 14 and 26 eyes)	Test: Received TTO (50%) twice-daily lid scrubs for 3 months and two doses of oral ivermectin (200 µg/kg, 1 week apart) (*n* = 15 eyes) Control: None	Improvement in symptoms (ocular surface inflammation such as congestion and corneal vascularization) after 3 months	Improvement in symptoms: Clinical improvement in sign and symptoms in all patients (no *p*-value is reported)	6 (Medium)
Tighe et al., 2013 [81], China	Case study	A blepharitis patient with *Demodex* (Age: 60 years, *n* = 1)	Test (*n* = 1): Received T4O lid scrub (Cliradex^®^ lid wipes) BID for 8 weeks Control: None	DMC after 8 weeks DER after 8 weeks Improvement in symptoms after 8 weeks	DMC: 0 (BL:22) DER (%): 100% (0/22) Improvement in symptoms: Marked resolution of symptoms and clearer lashes (no *p*-value is reported)	6 (Medium)
Wu et al., 2019 [87], Chania	Quasi-experimental	Meibomian gland dysfunction (MGD) patients with *Demodex* (Mean age: 60.5 ± 13.6 years, *n* = 38 with 76 eyes)	Test (*n* = 13, 26 eyes): Received both 0.02% flurometholone eye drops (anti-inflammatory) TID and TTO wipes (concertation not reported) BID for 4 weeks Control I (*n* = 13, 26 eyes): Received TTO wipes (concertation not reported) BID for 4 weeks Control II (*n* = 12, 24 eyes): Received 0.02% flurometholone eye drops TID for 4 weeks	DMC after 4 weeks Improvement in ocular symptoms (pain, redness, itching, burning/stinging and foreign body sensation) after 4 weeks (0–10 points) AEs occurrence (Day 29) (Only the outcomes with clinical significance are considered for this study)	DMC (mean ± SD): 0.5 ± 0.4 (BL:6.1 ± 4.8) in Test vs. 1.2 ± 1.5 (BL:6.7 ± 3.0) in TTO vs. 4.3 ± 2.7 (BL:5.6 ± 2.9) in Flurometholone groups (*p* < 0.01) Improvement in ocular symptoms score (mean ± SD): 3.3 ± 2.2 (BL:5.3 ± 2.0) in Test vs. 2.8 ± 2.0 (BL:4.3 ± 2.0) in TTO vs. 2.0 ± 2.2 (BL:4.3 ± 2.9) in Flurometholone groups (*p* = 0.0836) AEs: No AEs observed in all groups	9 (High)
Yam et al., 2014 [82], China	Case series	Recurrent chalazion patient with *Demodex* (Mean age: 39.1 ± 10.2 years, *n* = 30 with 48 eyes)	Test (*n* = 16, 31 eyes): Received TTO (50%) weekly lid scrub and 0.5 mL tea tree shampoo lid scrub BID for 3 weeks Control: None	Success/cure rate in preventing recurrent chalazion after 6 months follow-up AEs occurrence	Success/cure rate: 96.8% after treatment (*p* = 0.002) AEs: No AEs observed	10 (High)
Yin et al., 2021 [109], USA	Case study	An ocular Blepharitis patient with *Demodex* (Age: 72 years, *n* = 1)	Test (*n* = 1): Received T4O (Cliradex^®^) lid wipes (no frequency and duration of treatment reported) Control: None	DMC after 8 months OSDI score after 8 months	DMC: 0 (BL:31 mites) DER (%): 100% (2/2 eyes) OSDI score: 15 (BL:37) (no *p*-value is reported)	6 (Medium)
Zhong et al., 2021 [94], China	Quasi-experimental	Blepharitis patients with *Demodex* (Mean age: 47.4 ± 7.5 years in Test group; 46.6 ± 6.7 years in Control group, *n* = 56)	Test (*n* = 28, 56 eyes): Received meibomian glands comparison massage weekly followed by eyelid cleansing with cotton swab soaked with TTO (no concertation reported) daily for 2 months Control (*n* = 28, 56 eyes): Received meibomian glands comparison massage weekly followed by cotton swab soaked with normal saline eyelid cleansing daily for 2 months	DER after 2 months The OSDI score after 2 months	DER (%): 78.6% (44/56) in the Test vs. 10.7 % (2/56) in control groups (*p* < 0.001) OSDI score (Mean ± SD): 19.6 ± 4.2 (BL: 25.6 ± 6.8) in Test vs. 23.8 ± 5.2 (BL: 25.8 ± 6.9) Control groups (*p* < 0.001)	9 (High)

### 3.5. Scabies Mites

The five reviewed studies on the effects of TTO on scabies mites included two in vitro studies [57,119], a mixed in vitro/clinical study [118], and two clinical studies (RCT [86] and non-RCT [113]) (Table 3 and Table 4). The in vitro studies that tested the acaricidal activity of TTO and its components against scabies mites are presented in Table 3. Two studies [57,119] evaluated the effects of TTO (5–15%) in solution and lotion vehicles while the remaining study [118] compared the effect of TTO (5%) with T4O (2.1%), α-terpineol (0.1%), and 1,8-cineole (0.1%) solutions. The outcome variables were mite-lethal time (median) [57], both mite-lethal time (median) and mortality rate (%) [118], and mite mortality rate (%) [119]. The findings showed that TTO (5–10%) and T4O (2.1%) solutions eradicated the mites within 10–60 min (median), while TTO (15%) lotion showed a 100% mortality rate within 3 h. However, α-terpineol and 1,8-cineole solutions required 690 and 1020 min, respectively, to eradicate the mites. In sum, TTO and its main component (T4O) demonstrated a promising in vitro scabicidal effect with a 100% lethal effect within 0.2–3 h.

The RCT [86] compared the cure rates of TTO (5%) cream and a combination of TTO and permethrin (5% each) cream with permethrin (5%) cream in pediatric scabies patients (*n* = 72). The TTO (5%) cream demonstrated higher efficacy (54%) than the combination cream (20.8%) and the active control groups (16.7%) (*p* < 0.05). The study also reported a minor skin irritation associated with TTO use, although this was not statistically different from the combination and active control groups (*p* > 0.05). The two case studies (*n* = 1 each) [113,118] explored a combination of topical therapy (5% TTO in 25% benzyl benzoate lotion) with oral ivermectin in crusted scabies patients, with both showing a 100% mite eradication rate.

### 3.6. House Dust Mites 

Six in vitro studies reported the acaricidal activity of TTO and its components against house dust mites (Table 5). Five of these studies [60,65,68,80,116] evaluated the effects of TTO (5–100%) solutions, while the remaining one [83] compared the effect of T4O (40 µL/cm^2^), α-terpineol (40 µL/cm^2^), and 1,8-cineole (40 µL/cm^2^) solutions. Mite mortality rate (%) was the outcome variable evaluated in all studies. TTO (5–100%) and T4O (40 µL/cm^2^) demonstrated 80–100% mite mortality rate [60,65,68,83,116], while TTO (100%, 0.1µL/cm^2^) demonstrated a 10% mortality rate [80]. TTO (5–100%) and its main component (T4O, 40µL/cm^2^) generally demonstrated promising in vitro activity against house dust mites with an 80–100% mortality rate.

**Table 3 pharmaceutics-14-01587-t003:** Descriptive characteristics of included laboratory Scabies studies (*n* = 3).

Study Setting	Study Design	Method/Assay	Intervention	Outcome Measure(s)	Treatment Outcome(s)	Quality Score
Fang et al., 2016 [57], France	In vitro (*n* = 530 *S. scabiei* mites from pigs)	Direct contact and fumigation bioassays: direct application of test and control solutions on mites placed in Petri dishes in contact assay and placing mites at the bottom of Petri dishes covered with filter papers impregnated with the pure EOs in fumigation assay followed by stereomicroscopic examination of mites for 180 min in contact assay and 60 min in fumigation assay	Contact assay (*n* = 20 in each group) 10% and 5% of TTO, Clove oil (ClO), Palmarosa oil (PO), Geranium oil (GO), Lavender oil (LO), Manuka oil (MO), Bitter orange oil (BOO), Eucalyptus oil (EO), Japanese cedar oil (JCO) and Cade oil (CdO) Paraffin oil (Control) Fumigation assay (*n* = 10 in each group) 100μL of 100% of the above EOs Paraffin oil (Control)	Mite-lethal time: Duration from treatment to non-viability (absence of movement in the legs and the gut)	Median lethal time (LT_50_) ± SD: Contact assay (10% and 5%, respectively, No SD reported for CLO and PO) TTO (10.0 ± 6.0 and 30.0 ± 18.0 min), ClO (10.0 and 10.0), PO (10.0 and 10.0 ± 3.2), GO (10.0 ± 2.9 and 20.0 ± 7.0), LO (20.0 ± 6.6 and 35.0 ± 20.0), MO (30.0 ± 7.5 and 60.0 ± 24.0), BOO (20.0 ± 8.0 and 50.0 ± 33.0), EO (20.0 ± 16.0 and 150.0 ± 44.0), JCO (90.0 ± 42.0 and 180.0 ± 7.8) and CdO (no effect) vs. Control (no data reported) (*p* < 0.0001) Fumigation assay: TTO (4.0 ± 0.4 min), ClO (5.0), PO (7.0 ± 1.7), GO (5.0 ± 1.9), LO (5.0 ± 1.6), MO (23.0 ± 8.7), BOO (10.0 ± 5.4), EO (5.0 ± 0.3), JCO (10.0 ± 3.4) and CdO (> 60.0) vs. Control (no data reported) (*p* < 0.0001)	17 (Reliable without restriction)
Walton et al., 2000 [119], Australia	In vitro (*n* = 282 *S. scabiei* var. *hominis* mites)	Direct contact bioassays: placing the mites on test and control products contained in Petri dishes and microscopic examination of their non-viability for 180 min and up to a maximum of 22 h	TTO (15%) lotion (*n* = 21) Permethrin (5%) cream (*n* = 87) Benzyl benzoate (BB, 250 mg/mL or 25%) lotion (*n* = 26) Ivermectin (50–8000ng/g) paste (*n* = 86) Lindane (10 mg/g or 1%) lotion (*n* = 8) Neem seed oil (0.3–0.5% azadirachtins) spray (*n* = 22) Emulsifying ointment (BP88, Control, *n* = 32)	Mortality rate: Duration from treatment to non-viability (absence of all movement and peristalsis of the gut)	Mite mortality rate (%): Within 3 h: 100% for TTO, BB, lindane, and Ivermectin vs. Control (no data reported) (*p* < 0.05) After 3–18 h: 65% in Permethrin (*p* < 0.05) vs. 37% in Neem (*p* > 0.05) vs. 20% in Control After 18–22 h: 96% in Permethrin (*p* < 0.05) vs. 90% in Neem (*p* > 0.05) vs. 80% in Control	18 (Reliable without restriction)
Walton et al., 2004 [118], Australia	In vitro (*n* = 103 *S. scabiei* var. *hominis* mites)	Direct contact bioassays: placing the mites on test and control products contained in Petri dishes and microscopic examination of their viability for 180 min and up to a maximum of 22 h	TTO (5%) solution (*n* = 10) T4O (2.1%) solution (*n* = 10) α-Terpineol (0.15%) solution (*n* = 15) 1,8-Cineole (0.1%) solution (*n* = 14) Combination mixture (T4O, α-Terpineol and 1,8-Cineole) (*n* = 10) Permethrin (5%) cream (*n* = 9) Ivermectin (100µg/g) paste (*n* = 10) Emulsifying ointment (BP88, Control, *n* = 20)	Mite survival time: Duration from treatment to non-viability (absence of all movement and peristalsis of the gut) Proportion of non-viable mites after treatment	Mite survival time (Median): 60 min for TTO vs. 35 for T4O vs. 690 for α-Terpineol vs. 1020 for 1,8-Cineole vs. 20 for Combination vs. 120 for Permethrin vs. 150 for Ivermectin vs. 1260 for Control (*p* < 0.05 for all except α-Terpineol and 1,8-Cineole) Mortality rate (%): Within 3 h (approximation): 100% for TTO vs. 90% for T4O vs. 10% for α-Terpineol vs. 10% for 1,8-Cineole vs. 90% for combination vs. 80% for permethrin vs. 60% for Ivermectin vs. 0% for Control	18 (Reliable without restriction)

**Table 4 pharmaceutics-14-01587-t004:** Descriptive characteristics of included interventional/observational scabies studies (*n* = 3).

Study Setting	Study Design	Study Participant	Intervention Description	Outcome Measure(s)	Treatment Outcome(s)	Quality Score
Zulkarnain et al., 2019 [86], Indonesia	Randomized double blind controlled trial	Children with scabies (Mean age: 13.7 ± 1.3 years in TTO, 13.6 ± 1.2 in TTO + permethrin and 13.0 ± 1.0 in Permethrin groups, *n* = 72)	Test I (*n* = 24): Received TTO (5%) cream Test II (*n* = 24): Received a combination of TTO (5%) cream and permethrin (5%) cream Control (*n* = 24): Received permethrin (5%) cream (no clear report on frequency of administration)	Cure rate after 2 weeks AEs occurrence	Cure rate: 54.2% (13/24) in TTO group vs. 20.8% (5/24) in Combination group vs. 16.7% (4/25) in Permethrin group (*p* = 0.008) AEs: Minor irritation: Week 1: 0/24 in TTO group vs. 1/24 in Combination group vs. 1/24 in Permethrin group (*p* = 0.624); Week 2: 6/24 in TTO group vs. 10/24 in Combination group vs. 2/24 in Permethrin group (*p* = 0.07)	3 (High)
Currie et al., 2004 [113], Australia	Case study	Crusted scabies patient with mites resistant to oral ivermectin treatment (Age: 47 years, *n* = 1)	Test (*n* = 1): Received 11 doses of a combination of TTO (5 %) in benzyl benzoate (25%) topical therapy (lotion) for 1 month plus ivermectin therapy (18 mg/dose, 7 doses) Control: None	Mite eradication rate after 10 and 20 days	Mite eradication rate: 98% (98/100) eradicated after 10 days with 100% eradication after 20 days	8 (High)
Walton et al., 2004 [118], Australia	Case study	Crusted scabies patient (Age: 20 years, *n* = 1)	Test (*n* = 1): Received multiple doses (number of doses not reported) of topical TTO (5 %) in benzyl benzoate (25%) therapy (lotion) plus oral ivermectin Control: None	Mite eradication rate (no outcome end point reported)	Mite eradication rate: 100% eradication rate	6 (Medium)

**Table 5 pharmaceutics-14-01587-t005:** Descriptive characteristics of included laboratory house dust mite studies (*n* = 6, interventional study *n* = 0).

Study Setting	Study Design	Method/Assay	Intervention	Outcome Measure(s)	Treatment Outcome(s)	Quality Score
	In vitro (*n* = not reported, *Dermatophagoides farinae* mites)	Direct contact assays: spraying the mites placed onto discs of carpet lining in the base of the chamber at a rate of 10 mL/m^2^ (no diagnostic device is reported)	TTO (5%) solution spray Neem (5% cold-pressed oil) solution spray Imidacloprid (0.01%) solution spray Microencapsulated permethrin (1%) solution spray d-phenothrin (0.37%, Control) solution spray	Mortality rate (proportion of non-viable mites, no description on mortality assessment) after 24 h, 7 days, and 3 months	Mortality rate (%) after 24 h: 81% for TTO vs. 50% for Neem oil vs. 100% for Imidacloprid vs. 100% for Microencapsulated permethrin vs. 100% for d-phenothrin Mortality rate (%) after 7 days: no report for TTO vs. 92% for Neem oil vs. high mortality (no report) for Imidacloprid, Microencapsulated permethrin and d-phenothrin Mortality rate (%) after 3 months: 42% for TTO vs. 46.8% for Neem oil vs. 80% for Imidacloprid vs. 100% for Microencapsulated permethrin vs. 80% for d-phenothrin (no *p*-value is reported)	4 (Not assignable)
McDonald and Tovey, 1993 [116], Australia	In vitro (*n* = 350 house dust mites, no report on species type)	Direct contact assays: placing the mites in mesh capsules and immersing them in test products for 30 min followed by examination of their mobility after 12 h (no diagnostic device is reported)	100% of TTO (*n* = 50), Citronella oil (CO, *n* = 50), Eucalyptus oil (EO, *n* = 50), Spearmint oil (SO, *n* = 50), and Wintergreen oil (WO, *n* = 50) solutions Benzyl benzoate solution (BB, 0.8%), (*n* = 50) Tween-only solutions (0.8%, Control) (*n* = 50)	Mortality rate (proportion of non-viable mites, absence of mobility) after 30 min Rate of killing	Mortality rate (data obtained from graph): 98% for TTO vs. 100 % for BB vs. 100% for CO vs. 90% for EO vs. 88% for WO vs. 82% for SO vs. < 10% for Control (no *p*-value is reported) 98% for TTO vs. 100% for CO vs. 100 % for BB (*p* > 0.05) Rate of killing: TTO was the fastest acting EO killing 79% of mites in 10 min (no *p*-value is reported)	15 (Reliable without restriction)
Priestley et al., 1998 [65], UK	In vitro (*n* = 40 *D. pteronyssinus* mites)	Filter paper contact assays: placing the mites on suspending filter papers impregnated with test solutions and microscopic examination their mobility for 30 min and their mortality for 2 h	TTO (10%) solution (*n* = 10) Lavender oil (LO, 10%) solution (*n* = 10) Lemon oil (LeO, 10%) solution (*n* = 10) Ethanol (Control) (*n* = 10) solution	Mortality rate (proportion of non-viable mites, absence of movement when touched with forceps) after 2 h Proportion of non-moving mites, (without touching) after 30 min	Mortality rate: 100% for TTO vs. 87% for Lavender oil vs. 80% for Lemon oil vs. 0% for Control (no statistics reported) Proportion of non-moving mites: 100% for TTO vs. 87% for Lavender oil vs. 63 % for Lemon oil vs. 0% for Control (no *p*-value is reported)	14 (Reliable with restriction)
Rim and Jee, 2006 [80], South Korea	In vitro (*n* = 2429 *D. farinae* and *D. pteronyssinus* mites)	Filter paper contact assay: placing the mites on filter papers impregnated with test solutions placed at the bottom of Petri dishes and microscopic examination of mites after 5 min	0.1µL/cm^2^ of TTO (*n* = 307), Pennyroyal oil (*n* = 302), Ylang ylang oil (*n* = 312), Citronella oil (*n* = 297), Lemon Grass oil (*n* = 309), Rosemary oil (*n* = 309) solutions, Ethanol (Vehicle control, *n* = 306) solution and Permethrin (Active Control, *n* = 287, dosage form not indicated)	Mortality rate (Proportion of non-viable mites, absence of movement when touched with a pin) after 5 min of contact	Mortality rate: 10% for TTO vs. 100% for Pennyroyal vs. 98% for Ylang ylang vs. 0% for Citronella vs. 61% for Lemon Grass vs. 0% for Rosemary vs. 0% for Vehicle Control vs. 0% for Active Control (no *p*-value is reported)	14 (Not reliable)
Williamson et al., 2007 [68], UK	In vitro (*n* = 40 *D. pteronyssinus* mites)	Mite chamber assay: placing the mites on filter papers impregnated with test solutions horizontally suspended in a chamber and microscopic examination of their mobility for 30 min and mortality for 2 h	TTO (10%) solution (*n* = 10) Lavender oil (LO, 10%) solution (*n* = 10) Lemon oil (LeO, 10%) solution (*n* = 10) Ethanol (Control) solution (*n* = 10)	Mortality rate (Proportion of non-viable mites, absence of response to stroking with a paintbrush)	Mortality rate: 100 % for TTO vs. 87% for Lavender oil vs. 80% for Lemon oil vs. 0% for Control (no *p*-value is reported) Proportion of non-moving mites: TTO 100% for TTO vs. 87% for Lavender oil vs. 63% for Lemon oil vs. 0% for Control (no *p*-value is reported)	15 (Reliable without restriction)
Yang et al., 2013 [83], South Korea	In vitro (*n* = 30–40 house dust mites, *D. farinae* and *D. pteronyssinus*)	Filter paper direct contact assay: placing the mites on filter papers impregnated with test solutions placed at the bottom of Petri dishes and microscopic examination of their non-viability for 24 h	T4O (2.5–40µL/cm^2^) solution (*n* = 30–40) α-Terpineol (40µL/cm^2^) solution (*n* = 30–40) 1,8-Cineole (40µL/cm^2^) solution (*n* = 30–40) Benzyl benzoate (BB, 50μL, Active Control, dosage form not indicated) (30–40) Methanol (50 μL, Vehicle Control) solution (*n* = 30–40)	Mortality rate (absence of appendages movement when prodded with a pin) after 24 h Lethal dose (LD_50_) for T4O	Mortality rate (Mean ± SD) (D. farinae and D. pteronyssinus, respectively): 100% both for T4O (5µL) vs. 100% both for T4O (20µL) vs. 100% both for T4O (10µL) vs. 80 ± 0.5 and 85 ± 1.2 for T4O (5µL) vs. 40 ± 0.8 and 35 ± 0.8 for T4O (2.5µL) vs. 0% both for α-Terpineol vs. 30 ± 0.6 and 28 ± 1.3 for 1,8-Cineole (no report on the controls and *p*-value) Lethal dose (LD_50_, µL/cm^2^) (95% CI) (D. farinae and D. pteronyssinus, respectively): 3.9 (3.8–4.0) and 3.5 (3.4–3.6) for T4O vs. 7.8 (7.8–7.9) and 6.0 (5.9–6.0) for BB	16 (Reliable without restriction)

### 3.7. Insecticidal Effect of TTO and Its Components against Lice 

Of 15 studies involving lice, 10 were in vitro studies [31,53,54,55,66,67,68,84,95,115], one was a mixed in vitro/clinical study [103], and four were clinical studies (RCTs [110,111] and non-RCTs) [105,106] (Table 6 and Table 7).

Eleven in vitro studies tested the pediculicidal activity of TTO and its components against headlice (Table 6). Seven studies [31,53,54,67,68,84,115] evaluated the effects of TTO (1–100%) solutions alone and one study [55] compared TTO (1% and 10%) solutions with its components, such as T4O (1% and 10%), α-terpineol (1% and 10%), and γ-terpinene (1% and 10%) solutions. Two studies [95,103] tested shampoo (0.5% TTO, 0.8% thymol, and 0.2% paw paw extract) and solution (20–23% TTO, 13–17% lemongrass, 5.2% rosemary, 9.4–10.4% lavender, and 1% thymus oils) formulations and one study [66] compared T4O (100%) and α-terpineol (100%) solutions. Louse mortality rate (LMR, %) was evaluated in nine studies [31,53,54,55,67,68,95,103,115] and lethal time (LT_50_, minutes) in two studies [66,84]. Three studies also determined the ovicidal effect of TTO (1–8%) [54,67] and α-terpineol (2–5%) [66]. TTO (1–100%), T4O (10%), and α-terpineol (10%) recorded a LMR of 80–100% while TTO (100%), T4O (100%), and α-terpineol (100%) killed the lice within an average LT_50_ of 32–100 min. One study [55] also reported a relatively similar LMRs for T4O (1%) and α-terpineol (1%), 26 and 22%, respectively, as opposed to 0% for TTO (1%). Two studies [53,67] revealed that TTO dissolved in ethanol (93%), or water (94%) had a higher in vitro efficacy compared with using other carrier oils (i.e., coco or sunflower oil) (50%). The ovicidal rate of TTO (1–8%) ranged from 59–100%, while α-terpineol (2–5%) recorded ovicidal rate of 20–89%. TTO (1–8%) also demonstrated a higher ovicidal rate when solubilised in ethanol (83–100%) than in water (59%). In sum, TTO and its main components (T4O and α-terpineol) demonstrated promising in vitro pediculicidal and ovicidal efficacy with 100% lethal effects against lice within 2 h of application.

The clinical studies [103,105,106,110,111] involved 242 participants (*n* = 241 with headlice [103,106,110,111] and *n* = 1 with body lice [105]) with 224 in the RCT [110,111] and 18 in non-RCT studies [103,105,106]. Both RCTs [110,111] were active comparator-controlled, and both explored a lotion containing a combination of TTO (10% *w*/*v*) and lavender oil (LO, 1% *w*/*v*) as a test intervention. The non-RCTs were a cohort study exploring a shampoo formulation of TTO (0.5%), thymol (1.0%), and paw paw extract (0.5%) [103] and two case studies exploring a shampoo formulation (9% TTO, 7% anise oil, and 4% lemon oil) against headlice [106] and a TTO containing body wash (6% TTO, 8% cinnamon, 14% oregano, 40% lavender, 10% peppermint, 5% citronella, 7% orange, and 6% rosemary oils) against body lice [105]. There was no clinical study that studied the main components of TTO for louse treatment. 

The outcome variables included cure rate (% of louse-free participants), ovicidal efficacy, and occurrence of AEs. In the four studies reporting cure rate [103,105,106,110], TTO (0.5–10%)-containing formulations demonstrated 95.4–100% efficacy against lice infestations. One study [111] reported an ovicidal efficacy of 44.4% for TTO (10% *w*/*v*) and LO (1% *w*/*v*) lotion. Three studies assessed AEs, reporting either no AEs for the shampoo containing TTO (9%) in combination with other oils such as anise (7%), and lemon (4%) [106], or mild to moderate skin irritation for the TTO (10%) and LO (1%) lotion [110,111].

**Table 6 pharmaceutics-14-01587-t006:** Descriptive characteristics of included laboratory lice studies (*n* = 11).

Study Setting	Study Design	Method/Assay	Intervention	Outcome Measure(s)	Treatment Outcome(s)	Quality Score
Akkad et al., 2016 [31], Egypt	In vitro (*n* = 180 *P. humanus capitis*)	Direct contact bioassay: direct application of the test and control solutions on lice placed on filter papers in Petri dishes followed by exposing them with the solutions for 60 min and electron microscopic examination of their mortality for 60 min	*n* = 30 allocated in each group TTO (5%) headlice gel Ivermectin (1%) lotion Olive oil (extra virgin, 100%) Lemon juice (fresh, 100%) Licid lotion (0.6 g tetramethrin/2.4 g piperonyl butoxide (Active Control), Distilled water (Vehicle Control)	Louse mortality rate (LMR, %): from treatment to non-viability (absence of all vital signs and movement of antennae or legs)	LMR (%): 96.7% in TTO vs. 100% for Ivermectin vs. 100% for Lemon juice vs. 23.3% for Olive oil vs. 93.3% for Active Control vs. 0% for Vehicle Control (no *p*-value is reported)	17 (Reliable without restriction)
Candy et al., 2018 [53], France	In vitro (*n* = 180 *P. humanus capitis*)	Filter paper contact bioassay: placing the lice on filter papers-impregnated with test and control solutions placed at the bottom of Petri dishes followed by exposing them with the solutions for 30 min and stereo-microscopic examination of their mortality for 180 min	*n* = 30 allocated in each group 1.75 mg/cm^2^ of TTO, Wild bergamot oil (WBO), Clove oil (ClO) lavender oil (LO) and Yunnan verbena oil (YLO) solutions diluted either in Coconut or Sunflower oils Distilled water (Control)	LMR: from treatment to non-viability (absence of all vital signs and movement of antennae or legs)	(data reported only in figure) LMR (%): Coconut and sunflower, respectively ~45% and 35% for TTO vs. ~55% and 45% for WBO vs. ~97% and 95% for ClO vs. ~38% and 35% for LO vs. ~75% and 55% for YLO vs. 20% for Coconut vs. ~10% for Sunflower (*p* ≤ 0.05) sunflower (estimated from the graph and no statistics reported)	15 (Reliable without restriction)
Downs et al., 2000 [55], UK	In vitro (*n* = 917 *P. humanus capitis*)	Filter paper contact bioassay: placing the lice on filter papers impregnated with test and control solutions placed at the bottom of Petri dishes followed by exposing them with the products for 120 min and examination of their mortality after120 min (no diagnostic device is reported)	TTO (1% and 10%) solution (*n* = 131) T4O (1% and 10%) solution (*n* = 168) α-Terpeniol (1% and 10%) solution (*n* = 162) γ-Terpinene (1% and 10%) solution (*n* = 142) Copper oleate (1% and 10%) solution (*n* = 83) Tetralin (1% and 10%) solution (*n* = 151) No treatment (Control) (*n* = 80)	LMR: from treatment to non-viability (absence of all internal and external movement on tactile stimulation)	LMR (1%): 0% for TTO vs. 26.1% for T4O vs. 21.7% for α-Terpeniol vs. 0% for γ-Terpinene vs. 0% for Copper oleate vs. 25.7% for Tetralin vs. 0% for Control (*p* < 0.001 for all except Copper oleate) LMR (10%): 86.4% for TTO vs. 100% for T4O vs. 100% for α-Terpeniol vs. 57.4% for γ-Terpinene vs. 0% for Copper oleate vs. 100% for Tetralin vs. 0% for Control (*p* < 0.001 for all except Copper oleate)	14 (Reliable with restriction)
Heukelbach et al., 2008 [115], Australia	In vitro (*n* = 175 *P. humanus capitis*)	Direct contact bioassay: immersion of lice clasping hair strands in the test and control solutions for one minute and placing them on the filter papers in Petri dishes for 20 min and microscopic examination of their mortality for 180 min	*n* = 25 lice used in each group TTO (5%) (Tea Tree Head Lice Gel^®^) gel Ardusi leaf extract (20%) and Baibu root extract (20%) (Lice Blaster^®^) TTO (10%) and LO (1%) (Neutralice^®^) spray Neem seed extract (Praneem^®^ Repel^®^) shampoo Eucalyptus oil (10%, Moov^®^) product Baibu (5%) and coneflower (10%) foaming gel (Lysout^®^) Permethrin (1%, Active Control, Quellada^®^) No treatment Control	LMR: from treatment to non-viability (absence of any vital signs such as gut movement and movement of antennae or legs, with or without stimulation using forceps)	(Data reported only in figure and data for some of the products are estimated from the graph) LMR (%): 96% for TTO gel vs. 16.7% for TTO and LO spray vs. < 5% for Neem seed extract shampoo vs. < 5% for Baibu and coneflower foaming gel vs. ~15% for Eucalyptus oil product vs. ~15% for Ardusi and Baibu extract vs. 82.1% for Permethrin vs. < 5% for No treatment control after 180 min (*p* < 0.0001)	18 (Reliable without restriction)
McCage et al., 2002 [103], USA	In vitro (*n* = not reported, *P. humanus capitis*)	Direct contact bioassay: direct application of the test and control products on lice placed on filter papers in Petri dishes followed by exposing them with the products for 30 min and microscopic examination of their mortality for 120 min	(*n* for each group is not reported) Shampoo A (containing 0.5% TTO, 0.8% thymol and 0.2% Paw Paw extract) Shampoo B (containing 1.0% TTO, 1.5% thymol and 0.5% Paw Paw extract)	LMR: from treatment to non-viability (absence of antenna/claw/leg movement or stomach musculature contractions)	LMR (%): Shampoo B was more effective than Shampoo A (no data and *p*-value is reported)	10 (Not reliable)
Priestley et al., 2006 [66], UK	In vitro (*n* = not reported, *P. humanus* clothing lice)	Filter paper contact bioassays: placing the lice on filter papers-impregnated with test solutions placed at the bottom of Petri dishes followed by exposing both with the products for over 180 min (10 min for eggs) and examining their mortality For ovicidal test, immersing gauze with eggs attached in test solutions for 10mins and examination for their hatchability (no diagnostic device or viability assessment method is reported)	Pediculicide test: 600 μL of (+)-T4O, Pulegone, (−)-T4O, nerolidol, Thymol, α-Terpineol, Carvacrol, Linalool, Perillaldehyde, Geraniol, Citral, Carveol, Mentho, Geranyl acetate, Linalyl acetate solutions, no treatment control, solvent control (*n* and tested concentration/dilutions are not reported) Ovicidal test (≥300 eggs each): 2% and 5% of Carveol, Geraniol, Menthol, Nerolidol, α-Terpineol, Thymol, no treatment control, solvent control	Lethal time (LT_50_): from treatment to non-viability (absence of movement of limbs and gut, and failure to respond when the legs were stroked with forceps) Ovicidal (%) rate	Mean LT_50_ (data presented only in graph and no *p*-value is reported): LT_50_ < 50 min: (+)-T4O < Pulegone < (−)-T4O < Thymol; LT_50_ < 100mns: α-Terpineol < Carvacrol < Linalool < Perillaldehyde < Geraniol; LT_50_ < 350 min: Citral < Carveol < Mentho < Geranyl acetate < Linalyl acetate Ovicidal rate (%, at 2% and 5%, respectively) 100% and 100% for Nerolidol and Thymol vs. ~90% and ≥ 89% for Geraniol > ~ 65% and ≥89% for Carveol > ~20% and ≥89% for α-Terpineol > ~10% and ≥89% for Menthol > Citral (no data) > Citronellic acid (no data) > Linalool (no data) > (+)-T4O (no data)	14 (Reliable with restriction)
Veal 1996 [67], Iceland	In vitro (*n* = 240 P. *humanus capitis* and 1200–2400 eggs)	In vitro pediculicidal efficacy: immersion of lice and eggs in the test and control solutions for 10 s and placing them on the gauze in Petri dishes and examining their mortality after 17 h contact (no diagnostic device is reported)	*n* = 20 lice and 100–200 eggs used in each group Red thyme oil (RTO) plus Rosemary oils (RO) Mixture (Mix A, 1%) Peppermint oil (PO) plus Nutmeg oils (NO) Mixture (Mix B, 1%) TTO plus Cinnamon leaf oils (CLO) Mixture (Mix C, 1%) Individual oils: TTO (1%), Oregano oil (OrO, 1%), Aniseed oil (AO, 1%, CLO (1%), RTO, (1%) solutions Ethanol 40% solution (Control I) Water (Control II)	LMR: from treatment to non-viability (non-viability assessment is not reported) Louse Egg mortality rate	LMR and ovicidal rate (%): Mixtures (lice and eggs, respectively) vs. Ethanol: 87.3% and 39.4% for Mix A vs. 100% and 82.4% for Mix B vs. 100% and 96.2% for Mix C Individual oils (lice and eggs, respectively) vs. Water: 94.1% and 59.1% for TTO vs. 100% and 99.3% for OrO vs. 86% and 25.5% for AO vs. 94.1% and 59.1% for CLO vs. 100% and 50% for RTO Individual oils (lice and eggs, respectively) vs. Ethanol: 93.2% and 83.3% for TTO vs. 100% and 100% for OrO vs. 100% and 100% for AO vs. 100% and 100% for CLO vs. 83.9% and 92% for RTO (no *p*-value is reported, and the mortality data adjusted for control using Abbott’s correction)	15 (Reliable without restriction)
Williamson et al., 2007 [68], UK	In vitro (*n* = 40 *P. humanus*)	Filter paper contact bioassay: placing the lice on filter papers impregnated with test solutions placed at the bottom of Petri dishes followed by exposing them with the products for 210 min and microscopic examination of their mortality	TTO (10%) solution (*n* = 10) Lavender oil (LO, 10%) solution (*n* = 10) Lemon oil (LeO, 10%) solution (*n* = 10) Ethanol solution (Control) (*n* = 10)	LMR: from treatment to non-viability (absence of response to stroking with a paintbrush)	LMR (%, Mean ± SD): 90 ± 8% for TTO vs. 50 ± 14% for LO vs. 10 ± 8% for LeO vs. 10 ± 10% for Control (no *p*-value is reported)	13 (Reliable with restriction)
Yang et al., 2004 [84], South Korea	In vitro (*n* = 3420 *P. humanus capitis*)	Filter paper contact and fumigation assays: placing the lice on filter papers impregnated with test and control solutions placed at the bottom of Petri dishes followed by exposing them with the products for 300 min and examination of their mortality (no diagnostic device is reported)	0.25 mg/cm^2^ of TTO and other 53 plant EO solutions (*n* = 60) Acetone (Control I, *n* = 60) δ-Phenothrin (Control II, *n* = 60) Pyrethrum (Control III, *n* = 60)	Lethal time (LT50): from treatment to non-viability (absence of movement or exhibited lethargic response)	Mean LT_50_ (95%CI): 31.5 (30.11–32.98) mins for TTO vs. 23.1 (20.49–25.89) for δ-Phenothrin vs. 25.3 (22.14–28.55) for Pyrethrum vs. No mortality for Acetone (no *p*-value is reported)	16 (Reliable without restriction)

**Table 7 pharmaceutics-14-01587-t007:** Descriptive characteristics of included interventional/observational lice studies (*n* = 5).

	Study Design	Study Participants	Intervention Description	Outcome Measure(s)	Treatment Outcome(s)	Quality Score
Barker and Altman, 2010 [110], Australia	Randomised assessor-blind controlled trial	Individuals with headlice (*n* = 132, Age range: 4–12 years)	Test group I (*n* = 43): Received TTO (10% *w*/*v*) and lavender oil (LO, 1% *w*/*v*) (TTO/LO, NeutraLice^®^) lotion applied three times on Days 1, 7, and 14 Test group II (*n* = 45): Received Suffocation product (Benzyl alcohol, NeutraLice Advance^®^) applied three times on Days 1, 7, and 14 Control (*n* = 44): Received Pyrethrins (1.65 mg/g) and Piperonyl butoxide (16.5 mg/g) (P/PB, Banlice Mousse^®^ product) applied twice on Days 0 and 7	Cure rate (% of louse free participants) at Day 15 (test products) and Day 8 (Control) (AEs) occurrence	Cure rate: 95.4% (41/43) in TTO/LO group vs. 88.9% (40/45) in suffocation group vs. 22.7% (10/44) in Control group (*p* < 0.0001) AEs: 25 individuals with mild (*n* = 22) and moderate (*n* = 3) AEs) (*n* = 13 or 30.2% with stinging, *n* = 8 or 18.6% with flaky scalp/dry scalp and *n* = 4 or 9.3% with erythema among these, *n* = 3 moderate AEs (*n* = 1, stinging of the eyes; *n* = 1, stinging of the neck; and *n* = 1, skin erythema) in TTO/LO group vs. 3 (6.7%) individuals with mild AEs (flaky scalp/dry scalp) in Suffocation group vs. 4 (6.8%) individuals with mild AEs (flaky scalp/dry scalp and erythema, 1 subject (2.3%) in P/PB group	5 (High)
Barker and Altman, 2011 [111], Australia	Ex vivo Randomised assessor-blind controlled trial (ovicidal study)	Individuals with headlice (*n* = 92, Age range: 4–12 years)	Test group I (*n* = 31): Received TTO (10% *w*/*v*) and LO (1% *w*/*v*) (TTO/LO, NeutraLice^®^) lotion applied once on Day 1 Test group II (*n* = 31): Received Suffocation product (Benzyl alcohol, NeutraLice Advance^®^) applied once on Day 1 Control (*n* = 30): Received eucalyptus oil (11% *w*/*w*) and lemon tea tree oil (1% *w*/*w*) pediculicide (EO/LTTO, MOOV^®^) applied once on Day 1	Ovicidal rate (Per cent ovicidal efficacy) after 14 days AEs occurrence	Ovicidal rate (%) (SD): 44.4% (23%) in TTO/LO group vs. 68.3% (38%) in Suffocation group vs. 3.3% (16%) in EO/LTTO group (*p* < 0.0001) Aes: 4 (12.9%) individuals with mild Aes (*n* = 3 stinging and *n* = 1 redness) in TTO/LO group vs. 0% in Suffocation group vs. 6 (20%) individuals with mild Aes (*n* = 2 stinging and *n* = 4 redness) in EO/LTTO group	5 (High)
McCage et al., 2002 [103], USA	Cohort study	Individuals with headlice (*n* = 16, Age: not reported)	Test (*n* = 16): Received a shampoo formulation (containing 0.5% TTO, 1.0% thymol and 0.5% Paw Paw extract) applied three times eight days apart Control: None	Cure rate	Cure rate: 100% (16/16) (no *p*-value is reported)	6 (Medium)
Novelo, 2015 [105], USA (Patents)	Case study	Individuals with body lice (*n* = 1, Age: not reported)	Test (*n* = 1): Received a body wash (3.7 mL) containing TTO (6%), Cinnamon oil (8%), Oregano oil (14%), Lavender oil (40%), Peppermint oil (10%), Citronella oil (5%), Orange oil (7%), and Rosemary oil (6%) daily for two days Control: None	Cure rate	Cure rate: 100% (1/1)	3 (Low)
Whitledge, 2002 [106], USA	Case study	An individual with headlice (*n* = 1, Age: 8 years)	Test (*n* = 1): Received a shampoo containing TTO (9%), Anise oil (7%), Lemon oil (4%), SD alcohol (50%), water (28%), and fragrance (2%) applied once for 10–15 min Control: None	Cure rate AEs occurrence	Cure rate:100% (1/1) AEs: No reports of sensitivity or adverse reactions	7 (High)

### 3.8. Insecticidal Effect of TTO and Its Components against Fleas

Four studies (in vitro [95,104] and in vivo [96,105]) explored the insecticidal activity of TTO solution alone [96] and in combinations with other EOs [95,104,105] against dog and cat fleas (Table 8). We did not find studies exploring TTO and its components against human, rat, and sand fleas.

The in vitro studies evaluated the effects of TTO containing solution (20% TTO, 8% *Lippia javanica*, 13% lemongrass, 5.2% rosemary, 9.4 % lavender, and 1% thymus oils) [95] and aqueous formulation (1% TTO, 3.1% basil, 3.1% peppermint, 1.5% lavender, and 1.3% lemongrass oils) [104] while the in vivo studies assessed TTO (3%) solution alone [96] and TTO containing shampoo (6% TTO, 8% cinnamon, 14% oregano, 40% lavender, 10% peppermint, 5% citronella, 7% orange, and 6% rosemary oils) [105] on flea-infested cat and dogs. Flea mortality rate (%) and cure rate (%) plus improvement in local infection were the outcome variables evaluated in the in vitro [95,104] and in vivo studies [96,105], respectively. All the tested TTO (1–20%) containing formulations demonstrated a 100% in vitro mortality rate within 3–24 h and a 100% in vivo efficacy in 10 days (Table 8). One study [96] also revealed that TTO reduced local infections and promoted the healing of scratches associated with flea infestation. In sum, TTO (1–20%) showed promising in vitro and in vivo insecticidal activity against dog and cat fleas.

### 3.9. Safety and Treatment Satisfaction

In this review, AEs were assessed in 22 studies, with more than half of them [52,56,59,62,72,77,82,87,106,107,108,120] (all *Demodex* studies except one headlice study [106]) reporting no AEs and the remainder [74,79,86,92,97,98,99,110,111,121] (all *Demodex* studies except one scabies [86] and two headlice [110,111] studies) reporting mild to moderate skin irritations. The commonly reported AEs of TTO included skin irritation (burning, stinging, pruritus, and erythema), skin erosion, skin dryness, and skin rash for headlice and scabies, and ocular irritation for *Demodex* infections [74,79,86,92,97,98,99,110,111,121]. The studies assessing AEs, including the tested dosage forms and their treatment schedule, are summarised in the Appendix A, pp. 5–6).

Treatment satisfaction, compliance, or preference of TTO and its components were assessed only in five *Demodex* studies with reports of 100% satisfaction for eyelid wipes containing T4O (2.5%) and hyaluronic acid [121], 66% satisfaction, and 100% compliance for eyelid wipes containing T4O (2.5%) plus hyaluronic acid (0.2%, moisturiser) [73], 87% compliance for eyelid wipes containing T4O (0.1%) and sodium hyaluronate [69], and compliance of 72% for TTO (50%) lid scrub plus TTO (10%) shampoo [77]. TTO (0.02%) cleansing foam was also preferred (47%) to the other oral (33.3% for ivermectin and 5.2% for metronidazole) and topical counterparts (2.1% for daily lid hygiene, 7.3% for TTO 5% ointment, and 5.2% for metronidazole 2% ointment) [61].

### 3.10. Dosage Forms and Topical Pharmaceutical Formulations

Only topical pharmaceutical formulations containing TTO and its components were investigated in the included studies. Fourteen studies explored two (*n* = 13) [52,58,61,70,75,76,81,82,98,101,102,115,118] or three [97] different formulations, while the remainder (*n* = 62) explored a single topical formulation. Eyelid scrub (or sterile wipes) (*n* = 29) [58,59,63,69,72,73,75,76,77,78,79,81,82,87,88,89,90,91,92,94,97,98,101,102,107,108,109,120,121] was the most widely explored formulation followed by solution (diluted or undiluted) form of TTO and its components (*n* = 26) [53,54,55,57,60,64,65,66,67,68,70,71,80,81,83,84,95,96,97,100,104,105,112,114,116,118], shampoo (*n* = 11) [52,62,75,76,82,93,97,98,101,103,106], ointment (*n* = 6) [58,61,85,99,102,117], gel (*n* = 5) [31,52,56,74,115], lotion (*n* = 5) [110,111,113,118,119], cream (*n* = 1) [86], foam (*n* = 1) [61], and spray (*n* = 1) [115]. Most preclinical studies explored TTO and the components in diluted and undiluted solution forms. For clinical studies, TTO (50%) weekly and TTO (0.5 mL or 10%) daily scrub followed by TTO ointment in *Demodex* studies, and TTO lotion in scabies and headlice studies were the most widely investigated formulations.

**Table 8 pharmaceutics-14-01587-t008:** Descriptive characteristics of included laboratory flea studies (*n* = 4).

Study Setting	Study Design	Method/Assay	Intervention	Outcome Measure(s)	Treatment Outcome(s)	Quality Score
De Wolff, 2008 [95], USA	In vitro (*n* = 200 fleas (*Ctenocephalides felis*)	*Direct contact assay*: direct application of the test solutions to fleas placed on carpet and visual examination of their knockdown gently blowing on the carpet for their activity (after 1 h) and mortality (after 24 h) of the exposure (*no diagnostic device is reported for mortality assessment)*	TTO (20%), *Lippia* javanica (8%), Lemongrass (13%), Rosemary (5.2%), Lavender (9.4%), Thymus (1%) oils containing solution formulation (3g) No treatment controls	Flea Mortality rate after 24 h (*no viability assessment method is reported*) Knockdown rate after 1 h (absence of flea activity on blowing the carpet)	Flea Mortality rate: 100% Knockdown rate: 77.7% (*no p-value is reported, and mortality data adjusted for control mortality using Abbott’s formula*)	14 (Reliable with restriction)
Nair and Sasi, 2017 [104], USA	In vitro (*n* = 40 fleas (*C. felis*)	*Filter paper contact bioassays*: direct application of the test solutions on fleas placed on filter papers in Petri dishes and examination of their mortality for over 24 h after exposure (*no diagnostic device is reported for mortality assessment)*	TTO (1%), Basil (3.1%), Peppermint (3.1%), Lavender (1.5%), and Lemon grass (1.3%) oils containing aqueous solution formulation	Flea Mortality rate at 15, 30, 60, 120, and 180 min (*no viability assessment method is reported*)	Flea Mortality rate: 54% (in 15 min) vs. 75% (in 30mins) vs. 83% (in 60mins) vs. 93% (in 120 min) vs. 100% (in 180 min) (*no p-value is reported*)	11 (Not reliable)
Fitzjarrell, 1995 [96], USA	In vivo (cats and dogs (*n* = *not reported*) infested with fleas, *C. felis*)	The formulation rubbed into the fur of the flea infested animal and the animas were followed for 7 days	Test: Received solution formulation containing TTO (3%) applied on dog infested with flea every 2–3 days or once every 5–7 days Control: received no treatment	Cure rate (*no viability assessment method is reported*) Improvement in local infection (sores)	Cure rate: 100% in Test vs. no change in Control (*data not reported*) (*no p-value is reported*) Improvement in local infection: Reduced infection (sores) and healed wounds from scratches	14 (Not reliable)
Novelo, 2015 [105], USA	In vivo (cats (*n* = *not reported*) and dog (*n* = 1) infested with fleas, *C. felis*)	The formulation was applied on the fur of flea infested animals and followed for 8–10 days	Test: Received shampoo (3.7 mL) formulation containing TTO (6%), Cinnamon (8%), Oregano (14%), Lavender (40%), Peppermint (10%), Citronella (5%), Orange (7%), Rosemary (6%) oils every day for 8 days for the cat and once daily for 10 days for the dog	Cure rate (*no viability assessment method is reported*)	Cure rate: 100% for both cats and dogs (*no p-value is reported*)	17 (Reliable without restriction)

### 3.11. Quality Assessment

Among 30 in vitro studies, most (67%) of the studies [31,53,54,57,67,68,70,71,81,83,84,95,97,100,112,114,115,116,118,119] were regarded as reliable without restriction (Appendix A, pp. 12–15) indicating high methodological quality. One of the in vivo studies [105] was considered reliable with restriction, while the other [96] was graded as not reliable (Appendix A, pp. 15–16). The majority (69%) of RCTs [56,72,74,86,88,92,93,108,110,111,120] were graded as high quality for the Jadad scoring scale (Appendix A, pp. 7). The mean score for all RCTs was 3.4, indicating the overall high quality of the studies. Given one of the RCTs [107] had only a trial registry record, we did not report the result for its methodological quality assessment. The JBI tool assessment also revealed that most non-RCTs (66%) [58,59,69,73,75,76,78,79,82,87,89,90,94,97,98,99,101,106,113] had high methodological quality (Appendix A, pp. 8–11). The full assessment results of the studies are provided in the Appendix A (pp. 7–16).

## 4. Discussion

This is, to our knowledge, the first systematic review to rigorously assess all preclinical and clinical studies exploring the antiparasitic activity of TTO and its components against medically important ectoparasites. Our findings reveal several studies reporting promising acaricidal and insecticidal efficacy for TTO and its components. In addition, TTO and its components demonstrated significant improvement in ectoparasite-related symptoms.

### 4.1. Acaricidal Activity of TTO and Its Components against Mites

Mites are small arthropods of the family of Acarina. *Demodex*, scabies, and chiggers mites are the primary mite species of medical and/or public health importance [7].

Our review found that TTO and its components (mainly T4O) can completely eradicate *Demodex* mites and reduce mite-related symptoms without any serious AEs, as evidenced by multiple studies (Table 1 and Table 2). The efficacy reported in clinical studies was consistent with the in vitro results of laboratory-based studies and results reported for veterinary *Demodex* mites (*D. canis*), where the survival time of mites was 8–100 min for TTO (3.125–100%) [122]. *Demodex* mites (*Demodex folliculorum* and *D. brevis*) are the most common permanent ectoparasites in humans, infesting the pilosebaceous unit of the face and scalp skin to cause demodicosis [7,123]. They invade the base of the eyelashes, eyelash follicles, and sebaceous and meibomian glands, causing *Demodex* blepharitis (chronic ocular inflammation), cylindrical dandruff, disorders of the eyelash, meibomian gland dysfunction, lid margin inflammation, conjunctival inflammation, and corneal lesions [33,123,124]. Although there is limited evidence on *Demodex* mite–bacteria interactions, studies report that infestation with these mites causes bacterial infections either through transferring symbiotic bacteria living inside (e.g., *Bacillus oleronius*) or on the surface (e.g., *Streptococci* and *Staphylococci*) of the mites, or promoting bacterial invasion from the surrounding environment [33,125,126]. Currently, there is no standard drug treatment for demodicosis; however, existing treatment approaches include various oral (e.g., ivermectin and metronidazole) and topical (e.g., pilocarpine gel, metronidazole ointment, lindane lotion, permethrin cream, benzyl benzoate lotion, and TTO) treatments [28,30,40]. Topical (cream or eye drops) and oral antibiotics are also usually given together with the anti-*Demodex* drugs to treat the associated bacterial infections [125,127]. Among these treatments, TTO (T4O as a primary active ingredient) is considered the most promising treatment of *Demodex* blepharitis [30]. The promising preclinical and clinical findings for TTO and its components (5–50%) in this review further justify their current and future use as mainstay *Demodex* treatments. In addition, the antibacterial property of TTO holds tremendous potential in reducing the burden of bacteria associated with *Demodex* mite infestation compared with the currently used *Demodex* treatments. However, there is a lack of evidence on head-to-head comparisons of TTO and its components with currently used *Demodex* treatments, necessitating the need for well-designed studies to inform clinicians of the most efficacious and safe therapeutic options.

Although there are few preclinical and clinical studies assessing TTO against scabies mites, the in vitro evidence shows that TTO-based treatments alone or in combination with other agents (i.e., benzyl benzoate and permethrin) are more lethal to human scabies mites than standard scabies treatments. Equally promising activity was also reported in an animal study involving pigs infested with sarcoptic mange mites (*Sarcoptes scabiei* var. *suis*), with TTO (100%) killing 98.5% of the mites [128]. Scabies mites (*Sarcoptes scabiei* var. *hominis*) cause scabies in humans, a contagious parasitic skin disease affecting over 300 million people worldwide [129,130]. Scabies mites enter the body by burrowing into the skin [7]. As they burrow into the skin, they release antigens, including scabies mite inactivated protease paralogues (SMIPPs) and scabies mite serpins (SMSs) in their saliva and faecal matter which trigger inflammatory and immune (allergic) reactions towards the mites and their products [129]. The host rapidly develops intense itching and scratching causing skin abrasion or cracks [7,129,131]. The skin cracks serve as an entry point for pathogenic bacteria (e.g., *Staphylococcus aureus* and group A beta-haemolytic streptococci, [GAS])) leading to secondary bacterial infections, including potentially fatal systemic complications such as sepsis, post-streptococcus glomerulonephritis (APSGN), acute rheumatic fever (ARF), chronic kidney disease (CKD), and RHD [129,131]. The SMIPPs and SMSs are also suggested to contribute to the growth and survival of bacteria (e.g., in patients’ blood), possibly contributing to the potentially fatal disease sequalae [129,131]. Standard treatments for scabies include oral ivermectin, topical permethrin, and topical benzyl benzoate, and most of these treatments are potentially hazardous and associated with side effects, including severe skin irritation, headache, and nausea [130,132,133]. Furthermore, emerging drug-resistance of scabies mites is suggested as a critical failing of current treatments that demands the development of alternative scabies treatments [13,130,132]. Although additional RCTs are needed to confirm reported findings, the studies included in this review indicate promise for TTO-based formulations in the future of scabies treatment. Importantly, the results from a Phase II randomised controlled trial (*n* = 200) exploring the efficacy of TTO (5% *v*/*w*) gel in Australian Aboriginal settings are likely to provide additional insight into the utility of TTO for scabies treatment [130].

TTO showed significant in vitro activity against house dust mites, indicating TTO-based formulations could provide an effective control mechanism for these mites (Table 5). House dust mites (*Dermatophagoides farinae* and *D. pteronyssinus*) are a group of mites naturally associated with the dust and debris inside houses [9]. Although free-living, they are known to cause severe allergic diseases, including asthma, atopic dermatitis, and perennial rhinitis in humans [7,134]. Alongside meticulous hygiene, synthetic acaricides, such as benzyl benzoate, dibutyl phthalate, *N, N*-diethyl-meta-toluamide (DEET), and pirimiphos-methyl, have been used to control house dust mites [134]. However, similar to other treatments, these agents are associated with several drawbacks, including potent toxicity, damage to household contents, and widespread development of resistance of the mites against these treatments [83]. The promising in vitro findings for TTO warrant further well-designed and controlled studies with a head-to-head comparison of TTO with currently used treatments for house dust mites.

### 4.2. Insecticidal Activity of TTO and Its Components against Lice

Both preclinical and clinical studies considered in this review revealed promising ovicidal and pediculicidal activities for TTO against head and body lice. Several reports from laboratory and animal studies investigating TTO (1–20%) for the treatment of *Bovicola ocellatus* lice-infested donkeys [135,136,137,138] and *Bovicola ovis* lice-infested sheep [139,140] also showed promising efficacy with mortality rates in the range of 78–100%, verifying the findings from human studies. Three species of lice are known to parasitise humans: the head louse (*Pediculus humanus capitis*), the body louse (*P. humanus humanus*), and the crab or pubic louse (*Pthirus pubis*) [9,141]. Body lice are widely considered of public health importance because they transmit typhus fever, relapsing fever, and trench fever [141]. Infestation with body lice is associated with poor hygiene and precarious living conditions [141]. In contrast, headlice, the lice species most commonly found in humans, affect individuals irrespective of hygiene and living conditions [9,141]. Head lice prevalence is believed to be increasing steadily across the globe; while some extrapolate the annual occurrence to be hundreds of millions of cases, the Centres for Disease Control and Prevention (CDC) provides estimates of 6–12 million cases per annum in the USA alone [142]. Although there are no reported cases of human disease transmission linked with headlice [141], secondary bacterial infections (*Staphylococcus aureus* and GAS) can occur from constant scratching as a result of allergic reactions induced by lice saliva [141,143,144]. Several conventional pediculicides are currently available for pediculosis treatment [141]. However, their widespread use increased the development of resistant lice, driving the need for newer treatment alternatives with minimal potential for resistance [13,24,145,146]. The preclinical and clinical data from this review suggest that TTO is highly likely to be an effective lice treatment. Apart from its ovicidal and pediculicidal activities, TTO possesses good antibacterial and anti-inflammatory properties, offering additional benefits in potentially preventing the disease sequelae linked to secondary bacterial infections. In general, an effective lice treatment must possess activity against both lice and their eggs to break the parasite’s life cycle, with no requirement of repeated applications for drugs with additional ovicidal activity [146]. Given its pediculicidal and ovicidal effects, along with good antibacterial and anti-inflammatory activities and safety profile, it is reasonable to consider TTO, in a suitable pharmacological formulation, as a potential headlice treatment.

### 4.3. Insecticidal Activity of TTO and Its Components against Fleas

A few studies included in this review described promising insecticidal activity of TTO against cat and dog fleas. TTO was also found to have additional beneficial properties in reducing secondary infection and promoting the healing of scratches associated with flea infestation [96]. Fleas are small, wingless bloodsucking insects with a characteristic jumping movement [147]. The most important species are the rat flea (*Xenopsylla cheopis*), human flea (*Pulex irritans*), and cat flea (*Ctenocephalides felis*) [7,134]. A flea bite can lead to irritation, serious discomfort, and, most importantly, can be a means of pathogen transmission [148]. Rat fleas can cause plague and flea-borne typhus, while cat flea, the most abundant ectoparasite of cats and dogs, can cause cat scratch disease, flea allergic dermatitis, and tapeworm [134,148]. Sand fleas (*Tunga penetrans*) cause tungiasis, a WHO classified neglected epidermal parasitic skin disease, by burrowing into animal and human skin [149,150]. They secrete proteolytic enzymes to break the upper layer of the skin, which results in an inflammatory skin response by the host. As a result, patients develop intense itching and scratching that promotes the entry of pathogenic bacteria (such as *Streptococcus* sp. and *Staphylococcus* sp.) through the skin cracks [143,151]. Adult sand fleas frequently contain *Wolbachia* bacteria, which are known to infect many insect species [152]. Although the precise mechanism is yet to be determined, *Wolbachia* antigens are released following the death of the parasite, and these appear to play a key role in initiating severe localised inflammation commonly seen in patients with tungiasis [143]. Tungiasis inflicts pain and suffering on millions of people, particularly children with prevalence rate of up to 80%, living in sub-Saharan Africa, Latin America, and the Caribbean, leading to substantial human consequences, including childhood disability, stigma, and low quality of life [149,150]. Due to resistance development, many available flea treatments fail to eliminate flea infestation [134,148]. These treatments are also potentially hazardous to humans and must be applied by qualified personnel, requiring additional expense, which could be unaffordable to people living in resource-constrained settings [134]. For sand fleas, in particular, there is currently no proven, standard treatment. In desperation for relief, affected individuals physically extract the embedded flea using unhygienic sharp instruments, which can lead to severe inflammation and bacterial superinfections [153]. Other than several anecdotal or undocumented claims in tungiasis endemic settings, no study was found investigating TTO for tungiasis. TTO’s unique parasiticidal, antibacterial, and anti-inflammatory properties indicate its tremendous potential for reducing the severity of tungiasis and its complications. An exploratory tungiasis trial (ACTRN12619001610123) [154] has been planned to investigate the safety and efficacy of a TTO (5% *v*/*w*) gel formulation. Results from this investigation are likely to provide key evidence on the future place of TTO in the treatment of tungiasis.

### 4.4. Safety and Patient Compliance of TTO and Its Components 

Our review found no report of severe AEs or systemic reactions in the included clinical studies. Studies reported either no AEs or only mild to moderate skin irritation, suggesting the use of TTO and its components did not raise serious safety concerns. Multiple clinical studies investigating TTO against bacterial and fungal infections [155,156,157,158,159,160,161] also reported no or low risk of adverse skin reactions when TTO is formulated in a suitable pharmaceutical base at concentrations ≤25%. Regarding acceptance and compliance, multiple *Demodex* studies reported that treatments with TTO and its components were well-accepted, preferred over other available treatments, and there was good compliance by users [61,69,121]. Similarly, a report from a RCT [162] in children (mean age 6.3 + 5.1 years) with viral molluscum contagiosum demonstrated that TTO (75%) was well tolerated in the 30-day treatment period. From the preceding, TTO and its components appear to be sufficiently safe and acceptable to users to warrant further evaluation against these ectoparasites in well-designed RCTs.

### 4.5. Pharmaceutical Dosage Forms of TTO and Its Components

In addition to safety and efficacy, the nature of a pharmaceutical formulation and its ease of administration are crucial aspects to consider while devising a pharmacotherapy for ectoparasitic infections, because formulations can play a significant role in determining patient uptake. In this review, all but three in vitro studies [31,103,115] investigated diluted and undiluted solutions of TTO or its components. For *Demodex* mites, weekly eyelid scrubbing with TTO (50%) sterile wipes followed by daily eyelid scrubbing with TTO (5–10%) shampoo or ointment were the most explored treatments, and the two-time scrubbing practice seems to relate to the site of the *Demodex* mites, which usually reside at the base of the eyelashes and eyelash follicles [28]. The mechanical agitation from the weekly scrubbing is suggested to stimulate *Demodex* mites embedded inside the skin to move out to the surface and make it possible for the daily application to kill the mites before they start mating [97]. Innovative formulation designs involving nanoparticulate delivery mechanisms have demonstrated improved activity of TTO when compared to the conventional TTO formulations against bacteria. Such delivery methods could be valuable for *Demodex* treatment, as it can enhance the stability of TTO in the dosage form, control its release rate, and improve its penetration into the hair follicles [29,39,163,164]. Given that different formulations have the potential to influence the ocular exposure time and volume of product delivered to the eyelids [112], a comparative study of the most widely used anti-demodectic formulations is needed.

The reviewed scabies studies investigated lotion and cream formulations, while the headlice studies investigated solutions, shampoos, gels, and lotions/sprays. Heukelbach et al. (2008) [115] performed a head-to-head comparison of in vitro pediculicide efficacy of TTO (5%) gel and TTO (10%) and lavender oil (1%) lotion, reporting significantly higher efficacy for the gel (96%) compared with the lotion (17%). This finding is consistent with another in vitro pediculicide study [31] reporting similar efficacy (96%) for TTO (5%) gel formulation. Evidence [53,67,115] indicates that the presence of ethanol, as a solvent, in TTO formulation and enhanced skin partitioning from a lipophobic formulation base may contribute to superior results as opposed to involving solvents, such as acetone, or lipophilic carrier oils, such as coconut oil or sunflower oil. These findings suggest the need for formulation optimisation, and the likely impact it could have on bioactivity—such effects could include improved partitioning onto to the application site from the formulation base, prolonged skin contact time, and enhanced permeability of TTO into the parasite exoskeleton [67,115]. Gel formulations are generally most preferred for hair-bearing areas, such as the scalp [115,165]. They are likely to offer ease of application (less messy), better coverage, and enhanced skin partitioning and contact time, allowing the drug to permeate more effectively into the parasite [115,165]. In recent times, a new dimeticone gel-based formulation for headlice treatment has been developed to improve the formulation characteristics of the previous products (e.g., lotion) [166,167]. Similarly, a gel-based formulation may prove successful for TTO in ectoparasitic infestation treatment. In sum, there is a lack of head-to-head comparisons of different formulations for ectoparasite treatments, and further RCTs are required to inform efficacy, safety, and user preference.

### 4.6. Strengths and Limitations

This is the first systematic review to comprehensively analyse and summarise the antiparasitic activity of TTO and its components against mites, lice, and fleas to inform future researchers and clinicians. It rigorously assessed all preclinical and clinical studies exploring the acaricidal and insecticidal activity of TTO and its components. However, interpretation of the findings of this review should consider its limitations, including the narrative approach employed to review the available data. Heterogeneity in the study designs, their evaluation methods, outcome measures, and study periods precluded a meta-analysis. Most of the included clinical studies are non-RCTs (observational studies), limiting the quality and generalisability of reported findings. That said, well-designed observational studies are categorised as level II or III evidence, and they can still play an important role in informing RCTs in terms of hypothesis generation, refining research questions, and defining clinical conditions [168]. Different methodologies were used for some of the ectoparasites in the in vitro studies, suggesting the results from these studies should be explored in clinical practice with caution. However, from the methodological assessment results, most studies were found to be reliable enough to provide complementary evidence for the clinical efficacy results. Also, the studies sourced TTO from different providers in different countries, and the use of TTO with a low content of the main active components such as T4O, α-terpineol, γ-terpinene, and 1,8-cineole may have potentially had an impact on the study findings. To reduce the compositional variation from various factors, including the extraction methods, geographical locations, and harvest times [83], researchers should use oil that meets the International Standard ISO 4730 (“Oil of *Melaleuca*, terpinen-4-ol type”) [21].

## 5. Conclusions

The findings of this review show that TTO and its components are a promising treatment option for a range of ectoparasitic infections caused by mites, lice, and fleas. The compelling in vitro activity of TTO against ectoparasites has translated well into advanced investigations with promising outcomes observed in clinical studies, providing enough evidence to make recommendations for their clinical application. Also, most of the studies included in this review had high reliability and methodological quality. We found no study exploring TTO and its components against bed bugs, chigger mites (red bugs), and sand fleas. Given the promising activity of TTO and its components against similar ectoparasites, this review alerts researchers in this space to further explore the untapped potential use of TTO and its components as an alternative treatment against such parasites.

Ectoparasite infestations are usually associated with skin inflammation and secondary bacterial complications. Impetigo, a superficial paediatric bacterial infection, also occurs secondary to scabies in high-burden settings. Unresolved impetigo infections lead to serious sequelae with substantial morbidity and mortality, ranging from abscesses or bone infections and blood poisoning resulting in kidney and heart disease including acute post-streptococcal glomerulonephritis, acute rheumatic fever, and RHD. 

Considering the unique therapeutic attributes of TTO, such as antimicrobial, anti-pruritic, anti-inflammatory, and wound-healing effects, TTO could be an excellent alternative option to tackle neglected skin ectoparasitoses and associated bacterial and inflammatory complications, particularly in light of rapidly emerging global antimicrobial resistance crisis. This is particularly important in high-burden settings, where potentially fatal sequelae to ectoparasitoses arise from a complex interplay between environmental factors, bacterial pathogens, and skin parasites. TTO has been used widely over several decades with no evidence of resistance. The clinical decision on the use of TTO and its components against the discussed ectoparasites depends on multiple factors, such as efficacy, safety, duration of treatment, cost, ease of administration, and treatment acceptability. Further large-scale and high-quality RCTs can provide deeper insight into the therapeutic use of TTO for *Demodex*, head lice, and scabies infections. 

## Figures and Tables

**Figure 1 pharmaceutics-14-01587-f001:**
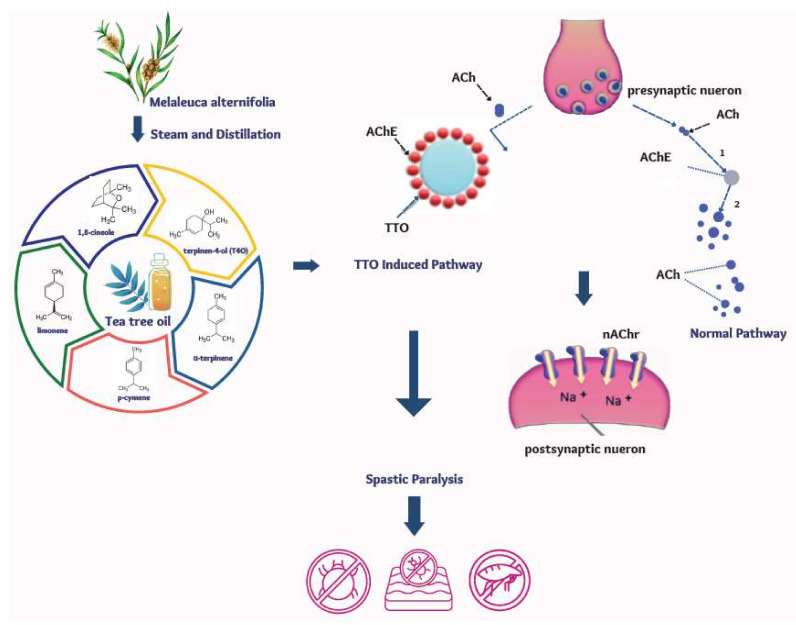
Antiparasitic activity of TTO attributed to its anticholinesterase activity (ACh: Acetylcholine; AChE: Acetylcholinesterase; nAChr: nicotinic acetylcholine receptors, and TTO: Tea tree oil, redrawn from Jankowska M. et al., 2018 [27]).

**Figure 2 pharmaceutics-14-01587-f002:**
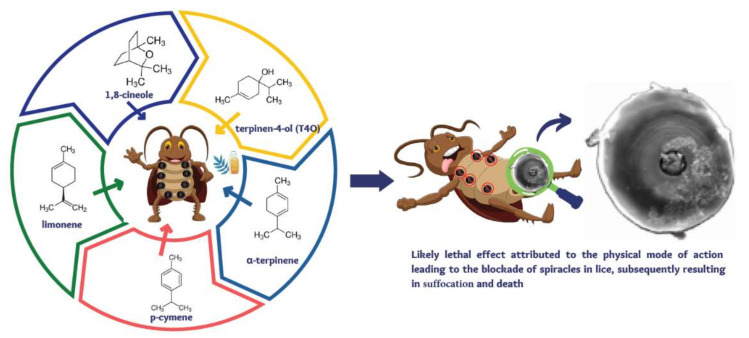
Mechanistic explanation of TTO’s pediculicidal activity (redrawn from Yingklang M. et al., 2022 [32]).

**Figure 3 pharmaceutics-14-01587-f003:**
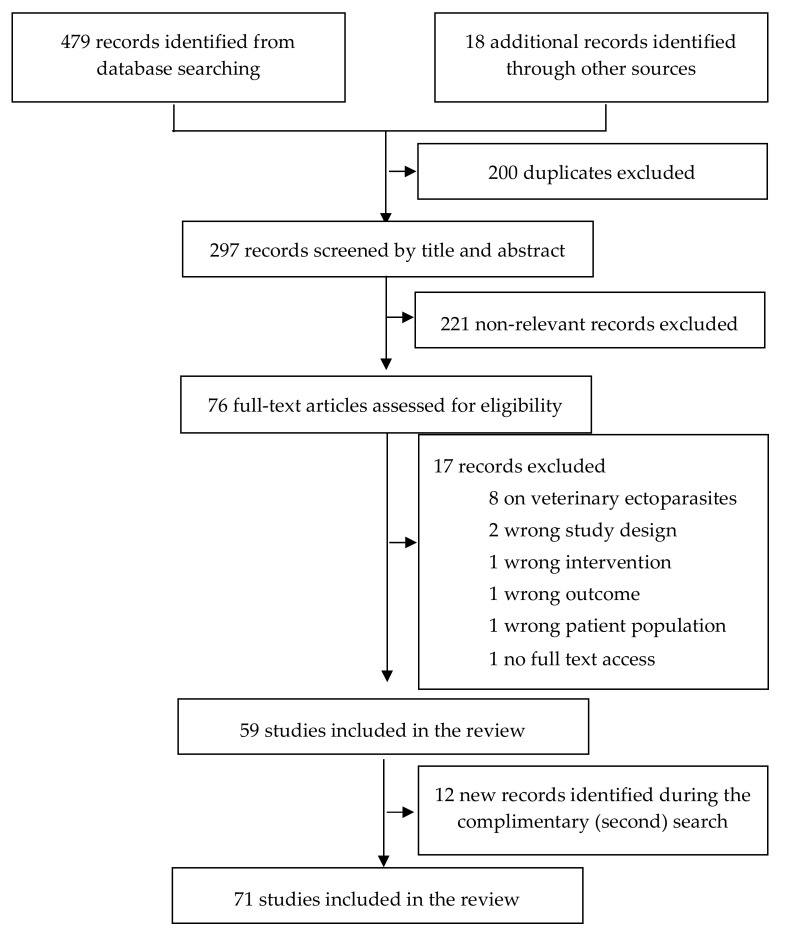
Study selection flow diagram.

## Data Availability

Not applicable.

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
