# Peer review of "Antiparasitic Activity of Tea Tree Oil (TTO) and Its Components against Medically Important Ectoparasites: A Systematic Review"

_pharmaceutics, 2022, doi:10.3390/pharmaceutics14081587_

Round 1

Reviewer 1 Report

This was an amazingly well organized and written review.  I have provided a few minor edits and suggestions as tracked changes within the attached PDF.

Your team should be proud of themselves for delivering such a clean and scholarly written article. Outstanding

Author Response

We would like to thank you for giving us this chance to revise our manuscript, and we appreciate the thorough review and constructive suggestions. We have carefully considered and addressed all the reviewers’ comments. We have explained how we revised the paper based on those comments and recommendations, citing the exact page and line numbers for the changes in the Author's Reply to the Review Report– please see the Author’s reply report for the reviewers’ comments uploaded with this submission. All the changes made in the manuscript following the reviewers’ comments are indicated in the revised manuscript with track changes.

We have now constructed two figures (Figures 1 and 2) illustrating TTO’s antiparasitic mechanism of action to address reviewer #1 feedback. We have provided the PDF files of the figures and the graphical abstract separately in this submission in line with the journal’s direction. Also, we have included the figure titles and captions in the introduction to indicate where the figures can be inserted. We have also provided a. We hope that these revisions have improved the paper’s framing and that the editorial team and reviewers find our responses satisfactory for publication.

Reviewer 2 Report

The present review entitled, ‘Antiparasitic activity of TTO and its components against medically important ectoparasites: a systematic review’ summarizes the preclinical (in vitro and in vivo) and clinical studies exploring tea tree oil (TTO) and its components against various ectoparasites, including mites (Demodex, scabies, house dust), lice, fleas, chiggers, and bed bugs. The idea behind the study is good and the design of this manuscript and writing is sound. Yet, the authors should clarify some major concerns before any possible consideration of this manuscript to be published in the journal of ‘Pharmaceutics’.

  • In #line_39-40 and 108, the authors have mentioned, “…..ectoparasites of medical importance.” What does the author mean by the term medically important ectoparasites? If the term has some significance to the study the author should explain this in the ‘Introduction’ section.
  • As authors have mentioned in the #Line_75-76, Why only a few compounds such as terpinen‐4‐ol (T4O), γ‐terpinene, α-terpinene, 1,8‐cineole, and terpinolene have a major significance in terms of bioactivity among hundreds of TTO compounds? The author should present the aspects of these compounds for better bioactivities.
  • The authors must produce some pictorial illustration of lethality (ways or mechanisms) caused to parasites by TTO compounds.
  • For a better understanding of the ‘Discussion’ part and to make this manuscript interesting, the authors should include some more figures explaining the acaricidal & insecticidal activity TTO and its components.
  • How the co-bacterial infection is prone to ectoparasite infections? The authors should explain the infection route and mechanism in detail.
  • The authors should clarify their perspective regarding the bottlenecks of the current studies and future challenges associated with the management of ectoparasites, in the ‘Prospect’ section.
  • Also improve the paper quality by referring the following papers https://www.sciencedirect.com/science/article/pii/S0300944020312212

Author Response

(The authors gave the same response as above.)

Reviewer 3 Report

The review is very well articulated, complete and thorough.
It is accompanied by a considerable bibliography, very useful for the reader who wants to deepen the single aspects dealt with.
My opinion is that the manuscript can be accepted for publication in "Pharmaceutics" in its current form.

Author Response

(The authors gave the same response as above.)

Reviewer 4 Report

Dear authors the manuscript is out of the aims of the Pharmaceutics, which include pharmaceutical formulation, process development, drug delivery, pharmacokinetics, biopharmaceutics, pharmacogenetics, and interdisciplinary research involving, but not limited to, engineering, biomedical sciences, and cell biology. The  manuscript has been submitted to a  Special Issue (Development of Novel Pharmaceuticals for the Treatment of Parasitic Diseases of the Section Physical Pharmacy and Formulation) but it does not fit also with the special issue and the session!  Tea tree oil is not a Pharmaceutical but it is an active principle. 

There are many  parts of the manuscript should be clarify before a possible resubmission

There are two well done reviews on tea tree oil activity on Demodex mites, which are published in Cochrane Database Syst Rev (2020 and 2019) and are only mentioned in the introduction but not included in results and the discussion sessions. These two papers have been used by the authors as models for the "search methods" session  and also figure 1 is quite similar (study flow diagram) . 

Main points  to be revised before a possible resubmission:

In vitro studies have a very low interest  especially  because no data could be reported concerning irritation, side effects etc... It is well known that tea tree oil is quite  toxic and  the use for oral administration is not permitted. It is well known that essential oils are generally very irritant after topical applications even when formulations  contains low percentages of essential oils.  Accordingly, the in vitro data should be not included in the manuscript.  

Second point is related to the formulations used in the studies, because it was the thematic of the special issue. The study should be more oriented to this part, which kind of formulations have been used and which ones are the best formulations  .  Data concerning also the  quality characteristics of the tea tree oil should be reported, if it was a standardized essential oil or simply which were the main components. 

In general, the manuscript seems to be a compilation of the available literature rather than a rational revision of the available  literature. Trial designs and formulations used are greatly varied and these aspects limit the analyses and comparison of the results. In addition, the effects of tea tree oil  may  strongly vary according to the severity of the manifestations. I strongly suggest to try to better select the criteria for the inclusion of the studies in the present manuscript in order to obtain a "systematic review", as reported in the title of the submitted manuscript. 

Author Response

(The authors gave the same response as above.)
